# Design, simulation, and experimental study of hydrostatic drive system for wide-span farming platform

**Jiaxiang Ma**[1,2], **Hongfeng Yan**[1,2,3]*, **Rongxuan Li**[1,2], **Falian Li**[1,2], **Xianfa Fang**[1,2]

**1** Chinese Academy of Agricultural Mechanization Sciences, Beijing, China, **2** National Key Laboratory of Agricultural Equipment Technology, Beijing, China, **3** Beijing Jinlun Kuntian Special Machine Co., Ltd., Beijing, China

* hongfeng216@163.com

**Data availability statement:** All relevant data are within the manuscript and its Supporting information files.

**Funding:** This research was funded by Research Project of China National Machinery Industry Corporation (ZDZX2022-1).

## Abstract

A versatile, robotic, and multi-functional wide-span farming platform, commonly known as a gantry tractor, can reduce soil compaction and enhance field production efficiency. In order to meet the functional requirements of wide-span farming platforms to achieve various steering modes in both transverse and longitudinal driving modes, while also improving the flexibility and stability of the platform's movement, a hydrostatic drive system was designed in this study based on an X-type dual-pump and four-motor system. The platform drive system adopts two variable pumps to drive four hydraulic motors, and the two pump–motor systems are distributed diagonally along the chassis. Each pump–motor system uses a flow diverter to prevent track slippage when the platform is driving. Mechanical analysis of the tracked platform while driving was first carried out. Based on the analysis results, the parameters of the main hydraulic components of the hydraulic drive system were calculated and selected. Then, the hydraulic drive system was modeled and simulated using the AMESim software. Finally, experiments were conducted under different driving conditions, including road tests in longitudinal mode and farmland tests in transverse mode. The experimental results showed that the hydraulic drive system exhibited a stable and synchronized performance. The average deviation rate of platform in longitudinal mode on cement roads was 5.0%. The deviation of the platform in transverse mode on dry and wet soil at the maximum driving speed were 6% and 2.1%, respectively. The test results demonstrated that the designed wide-span platform exhibited good driving stability, which validated the rationality of the design. These findings provide a reference for the improvement and optimization of wide-span farming platforms.

## Introduction

With the improvement of agricultural mechanization, field agricultural machinery and equipment are developing in the direction of having large-scale and heavy-duty characteristics, and the sizes of tires and the weight of equipment are increasing. As a result, the pressure on the

**Competing interests:** The authors have declared that no competing interests exist.

soil is on the rise [1,2], and the amount of research on soil compaction has begun to increase significantly [3]. Meanwhile, the impact of random traffic farming is gradually emerging, which makes the soil surface more susceptible to rainwater erosion, resulting in soil erosion, a poor soil water storage capacity, and more serious soil safety and environmental problems [4, 5]. Excessive soil compaction causes the soil structure to deteriorate. As compaction accumulates, the bulk density and strength of the soil increase, while the porosity decreases, thereby affecting the permeability to water, air, and crop roots in the soil [6]. The physicochemical properties and fertility traits of the soil are negatively affected [7]. Deep soil damage is difficult to repair, leading to energy waste and economic inefficiency. The high loads of tractors and other agricultural equipment is damaging to soil productivity, reducing the operational efficiency and affecting the crop yield. Therefore, it is urgent to reduce soil compaction.

To cope with such soil safety problems caused by the heavy lifting of large agricultural machinery and random traffic farming, controlled traffic farming (CTF) has gradually attracted attention [8–10]. CTF limits the area of soil compaction by agricultural machinery to as few permanent traffic tracks as possible [9], and the soil compaction caused by agricultural machinery operations such as tillage, sowing, fertilization, plant protection, harvesting, and transportation is limited to less than 10% of the cultivated land, while the traditional random traffic tillage operation mode accounts for 90% or more [11]. Compared with CTF, Wide-CTF features a significantly wider span between adjacent fixed traffic lanes, typically ranging from 8 to 12 meters, and sometimes more. Crop yield and comprehensive income will be further improved by Wide-CTF. However, most traditional agricultural machinery, such as tractors or self-propelled equipment, is not suitable for Wide-CTF. The wide-span farming platform, well-suited for Wide-CTF, effectively reduces soil compaction and increases crop yields in large farmlands, and thus, it has important application value and research significance.

The concept of a wide-span farming platform, also known as a gantry tractor, originated in the 1850s. Alexander Halkett [12] first constructed a steam cultivation machine called the Kensington Steam Cultivator, pioneering a permanent cultivation system fixed to the ground. The system was powered by a steam engine. Around the same time, Henry Grafton proposed a similar machine [13]. However, significant advancements in gantry tractor research were not made until the 1970s. David Dowler developed the first prototype gantry tractor, which featured a square-section frame that was approximately 12 m long, equipped with a 66 hp engine, a hydrostatic drive system, and castor and drive wheels that formed the tractor's main power train. Dowler's second prototype machine also utilized a hydrostatic drive, with each of the two drive wheels powered by independent gear pumps and motors, having a maximum speed of 19 km/h. The steering assistance structure for the driven wheels was optimized to enhance the steering performance.

The gantry tractor designed by David Dowler provided significant insights for the future development of wide-span platforms. Jim Taylor [14] utilized a wide frame vehicle called the Spanner to study soil compaction issues. The Spanner was first equipped with an advanced control system, enabling this four-wheel-drive, four-wheel-steering vehicle to operate in various movement modes. The Soil Dynamics Research Institute of the USDA Agricultural Research Service [15] organized the development of the Wide-Frame Tractive Vehicle. The Wide-Frame Tractive Vehicle featured a four-wheel hydrostatic drive, utilizing dual pumps to power four hydraulic motors for movement. It had an independent steering control system for each wheel and supported three steering modes: field mode, road mode, and pivot mode. Within these three modes, 12 different steering types were implemented. The AFRC Institute of Engineering Research [16,17] developed an experimental gantry that utilized two hydraulic motors to drive two crawler track units. In wide-span field mode, the gantry operated using

these crawler tracks. However, when driving on the road, the front and rear wheels needed to be lowered to the ground, and the gantry could only travel on roads when towed by a tractor, which limited its maneuverability. Due to persistent issues, such as being overweight, having poor maneuverability, and having complex drive systems, wide-span farming platforms have not achieved large-scale commercialization. To solve these problems, Dowler proposed an improved gantry design that retained four-wheel drive while simplifying the hydraulic system and reducing hydrostatic losses. The ASA-Lift company developed the ASA-Lift WS-9600, which enhanced the operation efficiency. Harvest CROO Robotics introduced an autonomous driving platform featuring a four-wheel steering mode with a zero turning radius, providing greater maneuverability with a GPS for unmanned operations. Kalverkamp Innovation GmbH developed the all-in-one system–NEXAT system, which is electrically driven, with generators currently powered by two independently driven 550 hp diesel engines. The design provides a lower, more-efficiently distributed weight, and it incorporates four-wheel independent steering and half-track-wheel assemblies for interchangeable wheel–track configurations. The NEXAT system is equipped with automated and intelligent functionalities, including autonomous driving, independent operation, and automatic implement docking.

Wide-span farming platforms, especially the NEXAT system, support the integration of modular implements for all agricultural tasks using wide-span transverse mode, including soil cultivation, planting, management, and harvesting. A comparison of a conventional tractor and a wide-span farming platform is shown in Fig 1. Unlike conventional tractors, whose implement widths often exceed road transportation standards, a wide-span farming platform can be driven longitudinally on roads or transversely in fields by converting between driving modes [18]. A wide-span farming platform has two driving modes: a transverse mode for field work and a longitudinal mode for road travel. In order to adapt to these two modes, the driving and steering system of the platform must be specially designed to ensure efficient travel and operation on both roads and fields. Based on these two basic modes, the platform can achieve omnidirectional movement. However, the performance of the platform in scenarios such as road travel, turning at field edges, and transitioning between field roads is constrained by the large-span beam structure. Consequently, the platform requires substantial traction and must maintain movement stability, both of which significantly affect its operation efficiency. Therefore, it is necessary to conduct research on the driving systems of wide-span farming platforms.

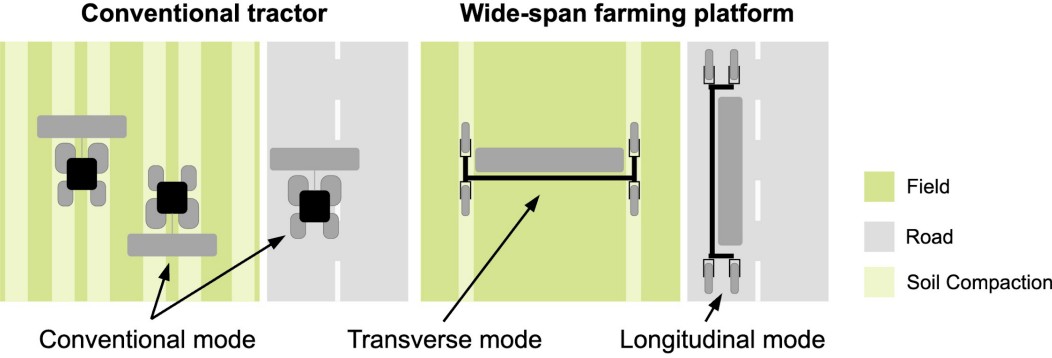

**Fig 1. Comparison of conventional tractor and wide-span farming platform.**

Overall, to meet the functional requirements for multiple steering modes in both the transverse and longitudinal driving modes of wide-span farming platforms, a hydrostatic drive system based on an X-type dual-pump and a four-hydraulic-motor configuration was designed. The AMESim software was used for simulation analysis of the hydraulic drive system to verify its feasibility. Finally, hydraulic system tests and travel deviation rate experiments were conducted to further validate the driving stability of the wide-span farming platform designed in this paper.

section*Design of wide-span farming platform and hydraulic drive system scheme

The structure of the wide-span farming platform is shown in Fig 2, which is mainly composed of a cab, chassis, implement module attachment system, front and rear support devices, suspension system, driving and steering system, operational power system, hydraulic system, and electrical system. The cab of the wide-span farming platform allows for the adjustment of its elevation and lateral rotation, enabling the driver to freely switch driving orientations based on the longitudinal or lateral operation requirements. This ensures a good driving view, accuracy, and comfort in the different operation modes. Overall dimensions of the wide-span farming platform is shown in Fig 3.

The chassis features a wide-span gantry-type eccentric structure, which forms a U-shaped space that facilitates the attachment of various modular implements. The chassis is composed of longitudinal beams, a front transverse beam, and a rear transverse beam. The longitudinal beams house the walking and operational power systems, and the outer sides are equipped

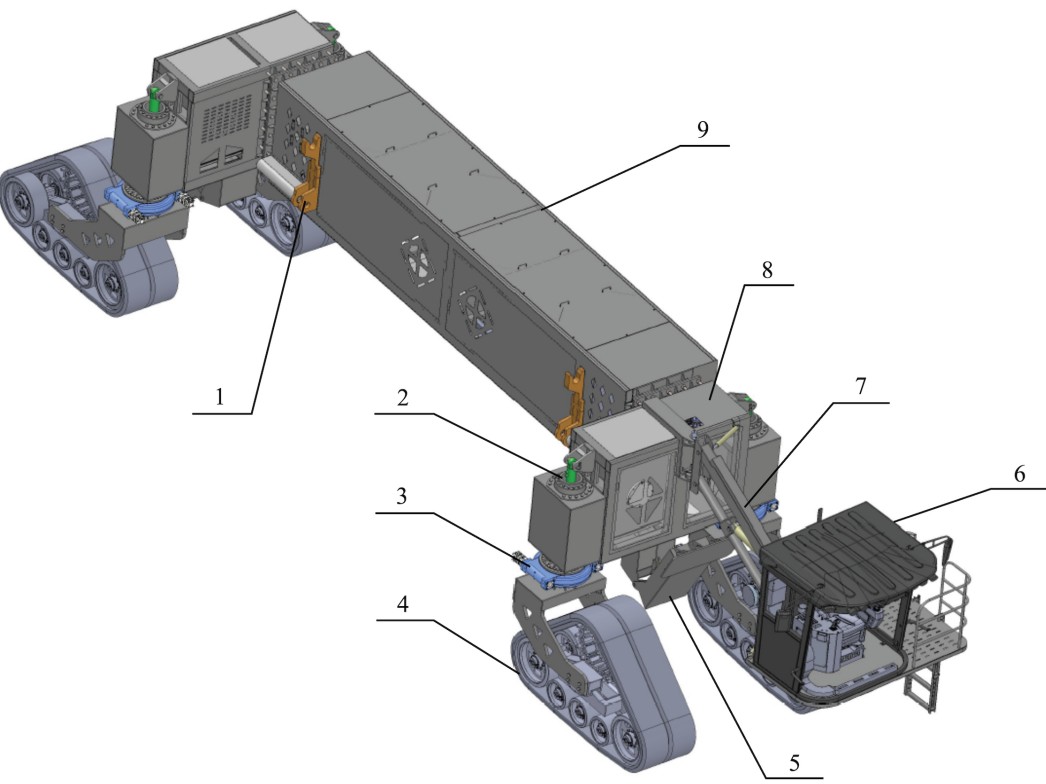

**Fig 2. Structure diagram of the wide-span farming platform.** 1—Implement module attachment device, 2—Suspension device, 3—Steering device, 4—Walking device, 5—Front and rear support devices, 6—Cab, 7—Cab position adjustment device, 8—Transverse beam, 9—Longitudinal beam.

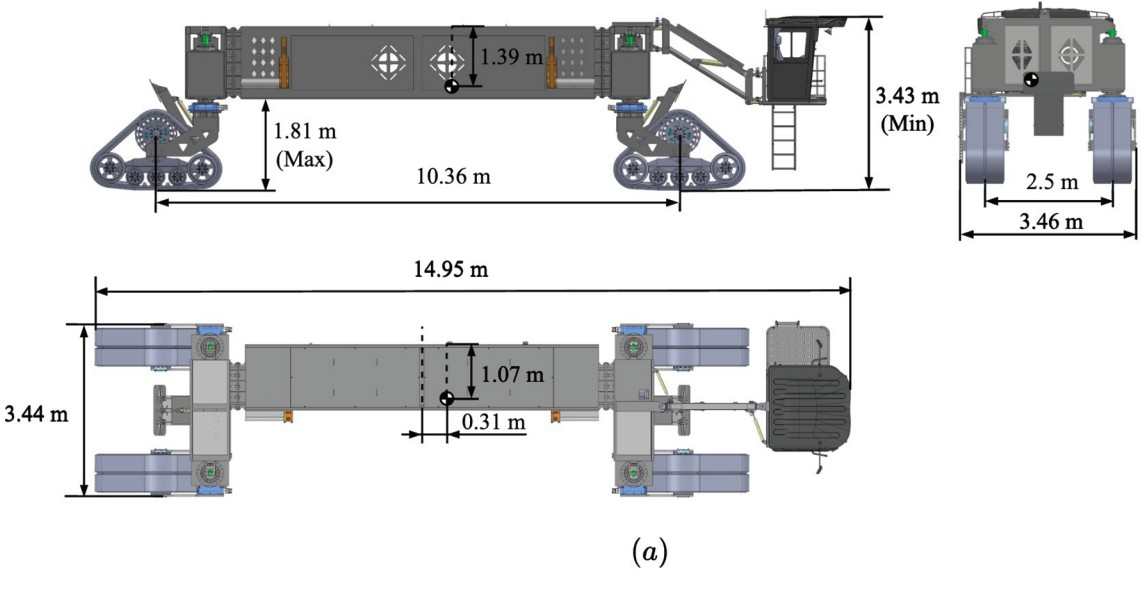

(a)

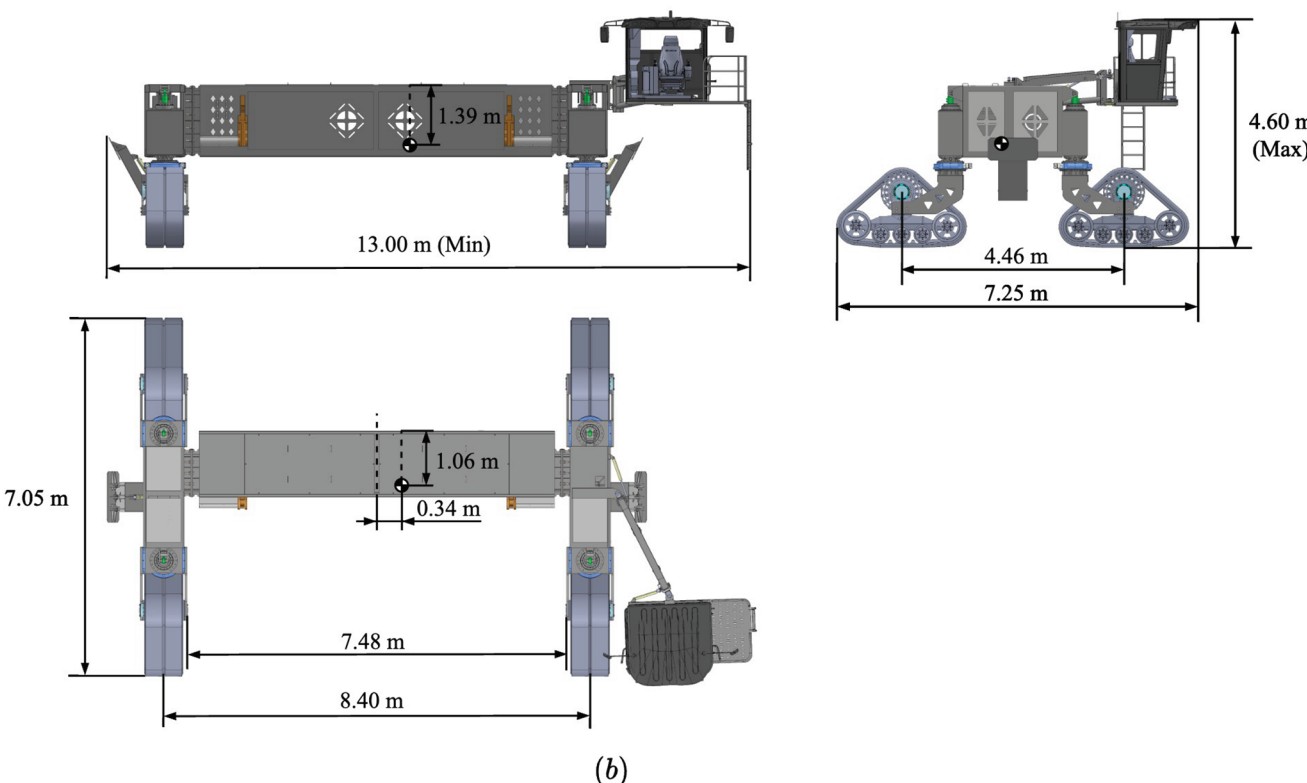

(b)

**Fig 3. Overall dimensions of the wide-span farming platform.** (a) Longitudinal mode. (b) Transverse mode.

with devices for attaching implements. The front and rear transverse beams have integrated suspension devices that contain a hydraulic oil tank, electronic control system, fuel tank, and hydraulic cooler.

The front and rear support devices enable mode switching between walking orientations. These devices are installed beneath the front and rear transverse beams and lift the wheel assemblies off the ground when switching between longitudinal and lateral driving modes. When switching between longitudinal and lateral driving modes on the spot, the front and rear support devices help reduce the steering resistance, offering protection for the structural components. The suspension system comprises four suspension devices located at the ends of the front and rear transverse beams. The suspension system works in conjunction with the front and rear support devices to complete the mode switching of the operating platform. The adaptive height adjustment of the platform improves its adaptability to various terrains during wide-span operations.

The walking power system, powered by the engine and hydraulic pump, provides hydraulic power for the entire machine. The distributed walking and steering system enables independent steering and drive control, which enhances the platform's maneuverability and flexibility. The steering modes of the tracked wide-span farming platform (longitudinal mode) are shown in Fig 4. The platform supports five steering modes: front-track steering (Fig 4b), rear-track steering (Fig 4c), four-track steering (Fig 4d), crab steering (Fig 4e), and in-place rotational steering (Fig 4f). The walking and steering system serves as the load-bearing component of the machine. Based on the large total mass of the machine and the soft soil in fields, to reduce soil compaction, a semi-tracked wheel system is adopted as the walking mechanism to decrease the ground pressure, enhance ground adaptability, and offer the interchangeable performances of rubber tires and semi-tracked wheels. The main design parameters of the wide-span farming platform are shown in Table 1.

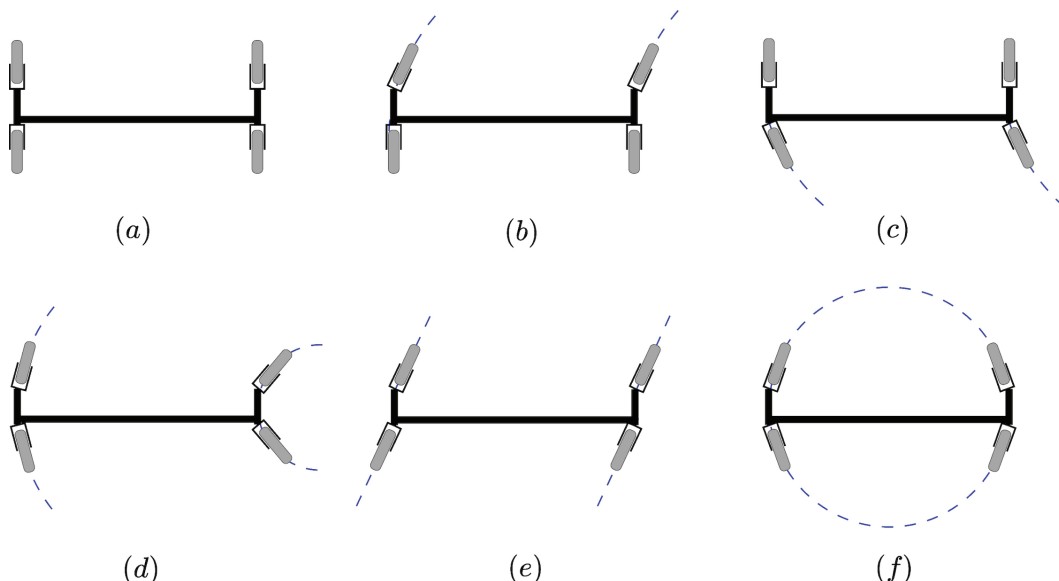

**Fig 4. Steering modes of the tracked wide-span farming platform (longitudinal mode).**

**Table 1. Main design parameters of the wide-span farming platform.**

| Parameters | Value | Remarks |
|---|---|---|
| Overall size* (mm) ($L \times W \times H$) | $14950 \times 3440 \times 3430$ | Longitudinal mode |
| | $13000 \times 7250 \times 4600$ | Transverse mode |
| Track span (mm) | 8400 | Transverse mode |
| Track base (mm) | 4460 | Transverse mode |
| Chassis mass (t) | 30 | Without implements |
| Maximum speed (km/h) | 18 | Road travel |
| | 10 | Field work |

* Lowest suspension position.

## Design of hydraulic drive system

The hydraulic drive system of the tracked wide-span farming platform should meet the demands of the longitudinal and transverse movement modes, as well as those of the in-place rotational steering mode. Considering that the platform must overcome significant steering resistance and provide a strong driving force, a hydrostatic drive system is used for transmission. The following design scheme for the hydraulic drive system of the tracked wide-span platform is proposed: the platform adopts an X-type dual-pump and four-motor system, where two closed-loop piston pumps drive four independently distributed low-speed, high-torque hydraulic motors using hydrostatic transmission [19]. Since the platform operates across large spans in the field, hydraulic transmission is chosen for its adaptability to unstructured working environments, allowing for stepless speed regulation and stable operation [20,21].

The X-type dual-pump and four-motor system is made up of two pump–motor systems, which are distributed diagonally along the chassis. This system is driven by a single engine powering two coaxial closed-loop variable-displacement pumps, each controlling two hydraulic motors distributed diagonally across the vehicle body. In normal driving and turning modes, the closed-loop hydraulic system automatically adjusts the differential distribution of motor speeds. The rotation control valve groups for the two hydraulic motors are composed of four two-way, two-position solenoid valves, controlling the direction of hydraulic oil flow through the walking hydraulic motors. By default, the position of the rotation control valve group for the walking hydraulic motors meets the walking requirements of the platform. The hydraulic oil flow and motor rotation directions for the hydraulic drive system in the longitudinal, transverse, and rotational modes are shown in Fig 5. The principle diagram of the hydraulic drive system of the wide-span platform is shown in Fig 6.

To prevent the issue of path deviation caused by the over-speed of individual motors due to uneven terrain during field operation and walking, a proportion diverter valve, which is a proportional directional valve element with flow sharing control, is installed. When activated, it can effectively control and lock the walking speed of each motor, enhancing the stability of the platform's straight-line travel. When the working platform is in front-track steering mode, rear-track steering mode, or rotational mode, the proportion diverter valve is closed, utilizing the self-adaptive flow characteristics of the hydraulic system to automatically allocate the flow to the same set of motors along the diagonal. The two-position four-way valve in the gear-shifting and braking valve assembly determines whether the parking-brake-control oil circuit is supplied with oil. When driving normally, the parking-brake-control oil circuit is not supplied with oil. When the solenoid valve is opened, hydraulic oil flows through the hydraulic motor parking device, enabling the parking function of the platform. The braking function

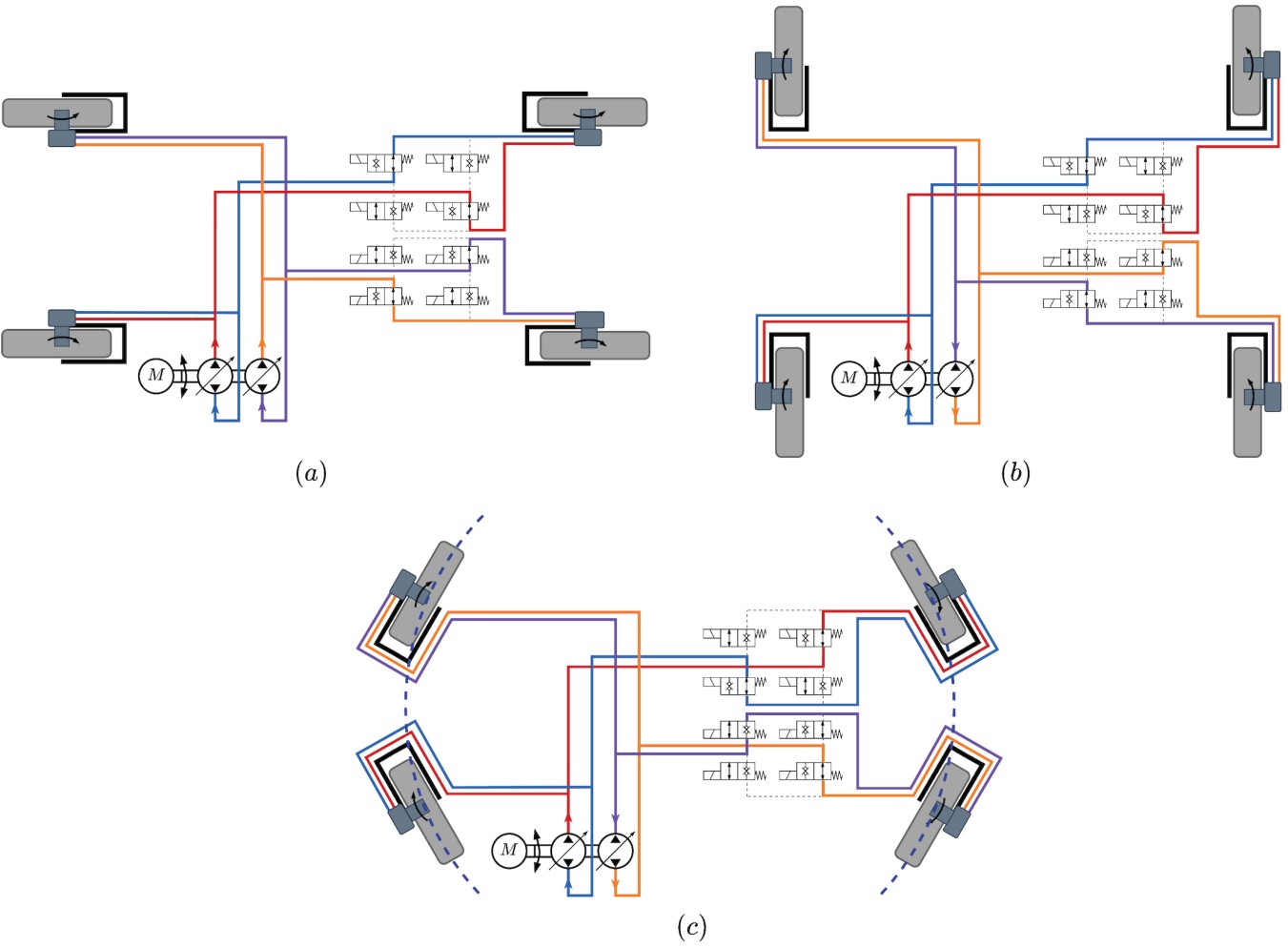

**Fig 5. Schematic diagram of hydraulic oil flow direction and hydraulic motor rotation direction.** (a) Longitudinal mode. (b) Transverse mode. (c) Rotational mode.

is achieved by controlling the pilot-type proportional sequence valve with an input electric signal, and the charge valve group provides backup oil pressure for braking.

## Calculation and selection of hydraulic system

**Analysis of driving characteristics.** When the wide-span farming platform operates in a field or traveling on the road, the hydraulic drive system drives the hydraulic motor to transfer the torque to the track-wheel assembly and drives the platform to move by overcoming ground resistance. The resistance generated by the interactions between the operation tools and the soil is not considered in this paper when the platform is traveling.

Based on the straight-line walking mechanics analysis of the tracked wide-span farming platform, as shown in Fig 7, the driving force $F_t$ provided by the hydraulic transmission system is balanced with the internal resistance $F_i$ of the platform and the sum of the external resistances acting on the platform. The internal resistance $F_i$ is the equivalent resistance of the power consumption within the platform system, which is the power consumed by the power device, transmission device, and track driving device when transmitting power from

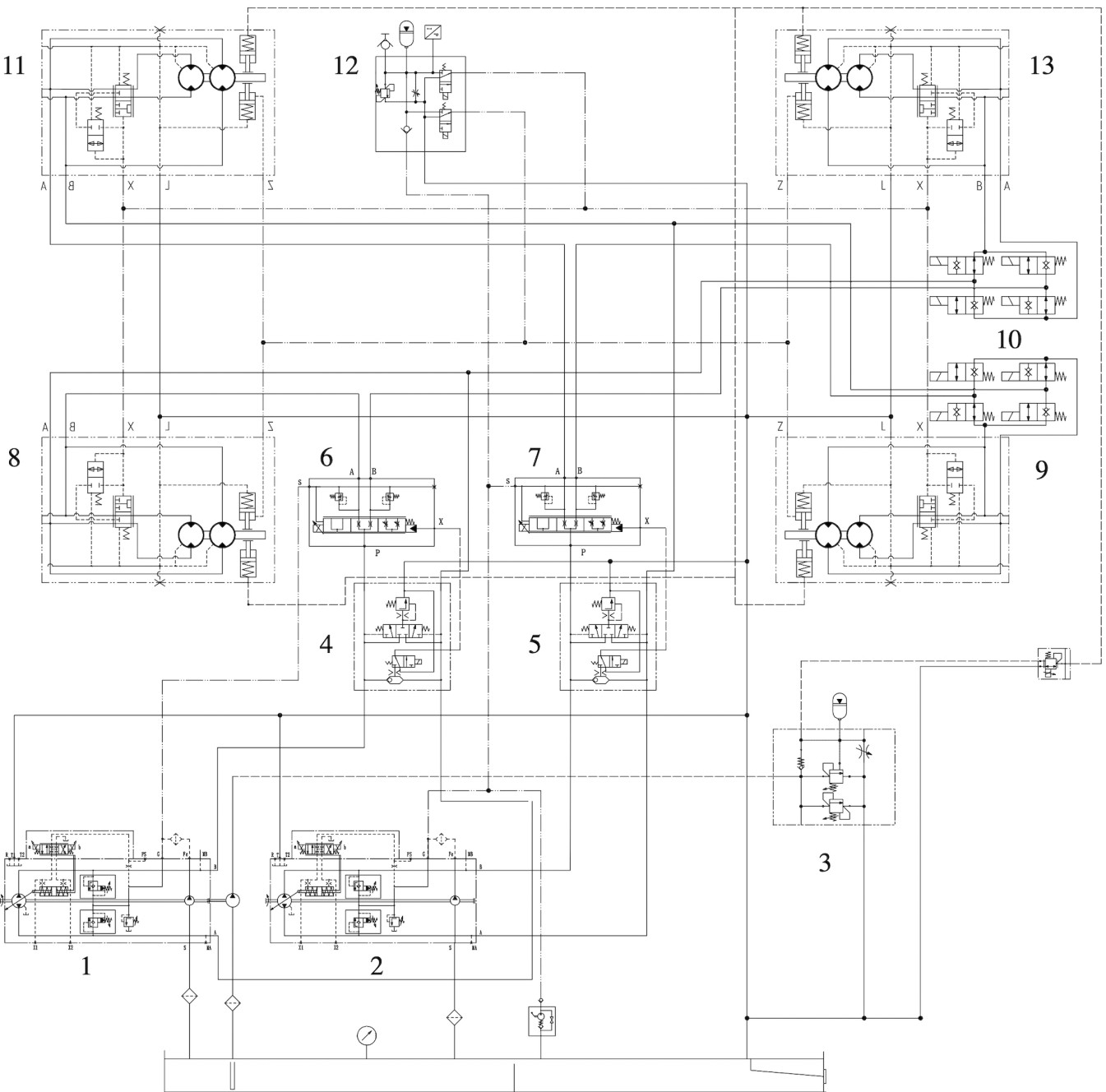

**Fig 6. Principle diagram of the hydraulic drive system of the wide-span platform.** 1—Variable-displacement pump, 2—Variable-displacement pump, 3—Charge valve group, 4—Flush valve group, 5—Flush valve group, 6—Proportion diverter valve, 7—Proportion diverter valve, 8—Right rear track hydraulic motor, 9—Right front track hydraulic motor, 10—Motor rotation control valve group, 11—Left rear track hydraulic motor, 12—Gear-shifting and braking valve assembly, 13—Left front track hydraulic motor.

the power source to the track. The external resistances acting on the platform include the ground deformation resistance $F_f$, downslope gravity force $F_s$, air resistance $F_w$, and inertial resistance $F_a$.

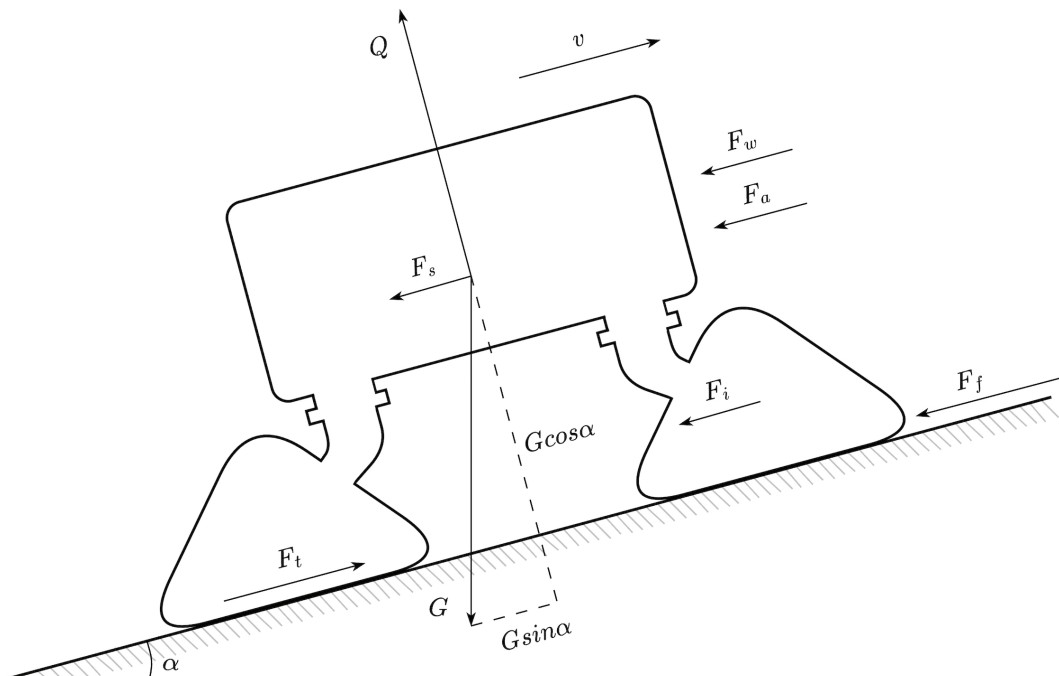

**Fig 7. Straight-line walking mechanics analysis of the tracked wide-span farming platform.**

Meanwhile, to ensure that the tracks of the platform do not slip, the maximum driving force $F_t$ must not exceed the adhesive force $F_\varphi$. The equation of the driving force for the wide-span farming platform can be expressed as

$$F_t = \sum F = F_i + F_f + F_s + F_a + F_w \tag{1}$$

$$F_t \leq F_\varphi \tag{2}$$

The tracks exert a force on the soil, causing deformation and generating a ground deformation resistance $F_f$. The downslope gravity force $F_s$ is the component of the weight of platform parallel to the slope. An inertial resistance $F_a$ is present when the platform is moving at a variable speed. When the wide-span farming platform moves laterally, it encounters air resistance $F_w$ from the moving direction, but when moving longitudinally, air resistance is negligible due to the smaller thrust surface of the platform. The formulas for the forces are as follows:

$$\begin{cases} F_i = \lambda_i G \\ F_f = fG \cos \alpha \\ F_s = G \sin \alpha \\ F_a = \delta ma \\ F_w = \dfrac{ACv^2}{21.15} \end{cases} \tag{3}$$

where $\lambda_i$ is the internal resistance proportional coefficient, G is the weight of the wide-span farming platform, $f$ is the ground deformation resistance coefficient, $\alpha$ is the slope angle,

$\delta$ is the mass increase coefficient, $m$ is the mass of the platform, $a$ is the acceleration of the platform, $A$ is the projected area of the platform, $C$ is the air resistance coefficient, and $v$ is the platform's driving speed without wind. The formulas and data are derived from *Driving Principle of Armored Vehicles*.

The adhesion force $F_\varphi$ is related to the total normal load $Q$ exerted on the ground by the platform and the adhesion coefficient $\varphi$. The adhesion force $F_\varphi$ is calculated as

$$F_\varphi = \varphi Q \tag{4}$$

Table 2 shows the values of the ground deformation resistance coefficient $f$ and adhesion coefficient $\varphi$ for the transverse and longitudinal modes of the platform in different environments. The data are derived from *Driving Principle of Armored Vehicles*.

Based on the operational scenarios in different modes, the corresponding design parameters were selected and substituted into the formulas. The calculation results are presented in Table 3. The maximum driving force $F_t$ of the platform was less than the adhesion force $F_\varphi$ in both the longitudinal road driving and transverse wide-span field operations. Therefore, the wide-span farming platform is able to obtain sufficient traction for movement.

**Parameter calculation and selection of key hydraulic components.** The value of $\eta_t$, $\eta_w$, $\eta_m$, $\eta_m v$, and $\eta_p v$ are derived from the *Hydraulic Engineer's Technical Manual*.

- Selection of Engine

The required engine power $P_D$ of the platform can be expressed as

$$P_D = \frac{F_t v}{3600 \eta_t} \tag{5}$$

where $\eta_t$ is the transmission efficiency of hydraulic drive system, with a value of 0.85. By substituting the values obtained from Table 3 into formula (5), the maximum driving power for flat road driving was calculated to be 176 kW, while the maximum power for field operations was calculated to be 180.6 kW. Therefore, the maximum power of the hydraulic transmission system occurred under field operation conditions, reaching 180.6 kW. The Cummins QSB 6.7 engine was selected with a maximum speed of 2200 r/min.

- Selection of Hydraulic Motor

**Table 2. Ground deformation resistance coefficient $f$ and adhesion coefficient $\varphi$.**

| Driving Mode | Pavement Type | $f$ | $\varphi$ |
|---|---|---|---|
| Longitudinal | Asphalt Road | 0.03–0.05 | 0.47–0.54 |
| | Concrete Road | 0.04–0.05 | 0.52–0.70 |
| | Dry Soil Road | 0.06–0.07 | 0.50–0.80 |
| Transverse | Grassland | 0.08–0.10 | 0.52–0.81 |
| | Muddy road | 0.10–0.15 | 0.35–0.62 |

**Table 3. Calculation results of driving force $F_t$ and adhesion force $F_\varphi$.**

| Driving Mode | $\alpha(°)$ | $f$ | $\varphi$ | $v$ (km/h) | $F_t$ (N) | $F_\varphi$ (N) |
|---|---|---|---|---|---|---|
| Longitudinal | 0° | 0.05 | 0.54 | 18 | $3.52 \times 10^4$ | $1.59 \times 10^5$ |
| | 10° | 0.05 | 0.54 | 8 | $8.60 \times 10^4$ | $1.56 \times 10^5$ |
| Transverse | 0° | 0.15 | 0.62 | 10 | $6.50 \times 10^4$ | $1.82 \times 10^5$ |

The wide-span farming platform adopts distributed independent drive technology, with each semi-tracked wheel group unit driven by an independent hydraulic motor. When the platform moves straight, its maximum driving force is equal to the sum of the traction forces of the four tracked wheel groups. Assuming that each tracked wheel group experiences the same force state, the maximum output torque of each track hydraulic motor can be expressed as

$$M_m = \frac{F_t R_w}{n \eta_w} \tag{6}$$

where $R_w$ is the radius of the triangular track's driving force, with a value of 0.405 m, $n$ is the number of track hydraulic motors, with a value of 4, and $\eta_w$ is the transmission efficiency of the triangular track wheel, taken as 0.92. The theoretical displacement of the track hydraulic motor is given by

$$V_m = \frac{2\pi M_m}{\Delta p_m \eta_m} \tag{7}$$

where $\Delta p_m$ is the pressure difference between the inlet and outlet of the walking hydraulic motor, with a value of 38 MPa, and $\eta_m$ is the mechanical efficiency of the walking hydraulic motor, taken as 0.92. The maximum rotational speed of the track hydraulic motor is

$$n_{\max} = \frac{1000\, v_{\max}}{60 \cdot 2\pi R_s} \tag{8}$$

where $v_{\max}$ is the maximum driving speed of the platform, and $R_s$ is the rolling radius of the triangular track, with a value of 0.522 m.

The parameters for the different driving modes were substituted into the above equations, and the calculated results of the track hydraulic motor are shown in Table 4. Finally, the MS18.2 dual-displacement motor from Shanghai Butuo Transmission Systems Co., Ltd. was selected, with the main technical parameters listed in Table 5.

For the wide-span farming platform proposed in this paper, in the longitudinal road driving scenario, while driving on a smooth road, a high gear is engaged, and the track hydraulic motor switches to half-displacement mode, delivering a low torque and a high

Table 4. Calculated results of track hydraulic motor.

| Mode | Gear | $v_{\max}$ (km/h) | $M_m$ (N·m) | $V_m$ (mL/r) | $n_{\max}$ (r/min) |
|---|---|---|---|---|---|
| Longitudinal | High | 18 | 3874 | 696 | 92 |
| | Low | 8 | 3874 | 9465 | 41 |
| Transverse | Low | 10 | 7154 | 1285 | 51 |

Table 5. Main technical parameters of the dual-displacement track hydraulic motor.

| Characteristic | Parameter | |
|---|---|---|
| | Full Displacement | Half Displacement |
| Displacement (mL/r) | 2099 | 1050 |
| Maximum speed (r/min) | 100 | 125 |
| Maximum torque (N·m) | 15000 | 7500 |
| Maximum pressure (MPa) | 45 | |
| Static braking torque (N·m) | 12000 | |

speed. When the platform is climbing a slope with the low gear engaged, the track hydraulic motor switches to full-displacement mode, delivering high torque. In the transverse field farming scenario, the platform operates in a low gear, with the track hydraulic motor in full-displacement mode, delivering a high torque.

- Selection of Hydraulic Pump

The output flow of the required hydraulic pump is determined based on the maximum input flow of the track hydraulic motor. Subsequently, the maximum displacement of the hydraulic pump is determined, and the model of the hydraulic pump is selected while meeting the maximum working pressure requirement. During the oil supply process from the hydraulic pump to the track hydraulic motor in the closed hydraulic system, there exists a pressure loss $\Delta p$ of the hydraulic oil in the system pipelines. Therefore, the maximum working pressure $p_p$ of the hydraulic pump satisfies the following conditions:

$$p_p \geq \Delta p_m + \Delta p \tag{9}$$

With $\Delta p$ set to 2 MPa, the maximum working pressure $p_p$ of the hydraulic pump should not be less than 40 MPa.

When the platform is in a low gear and working in the field at maximum speed, the input flow rate to the track hydraulic motor is at its maximum. The maximum input flow rate of the track hydraulic motor is calculated by the following equation:

$$Q_m = \frac{V_0 n_{\max}}{1000\eta_{mv}} \tag{10}$$

where $V_0$ is the full displacement of the track hydraulic motor, $n_{\max}$ is the maximum rotational speed of the track hydraulic motor, and $\eta_{mv}$ is the volumetric efficiency of the track hydraulic motor, taken as 0.95.

The hydraulic drive system of the platform adopts a dual-pump configuration, each driving two track hydraulic motors. If the leakage in the closed hydraulic system is negligible, the output flow rate of each hydraulic pump should be equal to the sum of the maximum input flow rates of the two track hydraulic motors. Therefore, the displacement of the hydraulic pump is

$$V_p = \frac{1000 \cdot 2Q_m}{n_p \eta_{pv}} \tag{11}$$

where $n_p$ is the rotational speed of the hydraulic pump, and $\eta_{pv}$ is the volumetric efficiency of the hydraulic pump, taken as 0.98.

The maximum output flow rate of the hydraulic pump was calculated to be 225.4 L/min. The engine speed was equal to the hydraulic pump speed, with a value of 2200 r/min, which resulted in a maximum displacement of 104.5 mL/r for the hydraulic pump. Finally, the HPV165 variable-displacement piston pump from Haitai Hydraulic Co., Ltd. was selected. The main technical parameters of the variable-displacement pump are shown in Table 6.

## Simulation of hydraulic drive system

The wide-span farming platform is capable of two driving modes: longitudinal road driving and transverse field farming. Each mode corresponds to different operational scenarios, with different requirements for the platform's driving performance. To verify the feasibility

**Table 6. Main technical parameters of the variable-displacement pump.**

| Model | Maximum Displacement (mL/r) | Maximum Speed (r/min) | Maximum Pressure (MPa) | Maximum Continuous Power (kW) |
|---|---|---|---|---|
| HPV165T-02 | 165.6 | 2500 | 42 | 173 |

of the hydraulic drive system of the wide-span farming platform, a simulation model of the hydraulic drive system of the wide-span farming platform was established in AMESim based on the working principle of the aforementioned hydraulic system and the parameters of the key components. The changes in the hydraulic characteristics of the driving system under different working conditions were analyzed via simulations.

## Establishment of hydraulic drive system simulation model

Based on the principles of the hydraulic drive system, a simulation model of the walking hydraulic system was established in AMESim, as shown in Fig 8. The simulation model primarily included hydraulic components, mechanical components, and control signals. The hydraulic components included bidirectional variable pumps, unidirectional fixed-displacement pumps, bidirectional fixed-displacement motors, a hydraulic tank, and various hydraulic valves, all of which were connected through hydraulic pipelines. The dual-displacement hydraulic motor was simplified into two parallel fixed-displacement hydraulic motors, and control signals were used to control the engine speed, hydraulic pump displacement, and opening or direction of each directional valve. Electrical signals were used to control the displacement of the variable pump, and a transfer function was used to match the input electrical signals with the displacement characteristics curve of the variable pump. The model parameters were determined based on the key component parameters provided above, and the main simulation parameters of the hydraulic system model are listed in Table 7.

During platform movement, the driver can select and switch gears based on different driving environments, changing the displacement of the track hydraulic motor to adjust the platform's maximum driving force and maximum speed. Therefore, the hydraulic drive system of the platform was simulated for three conditions: high-gear steady road driving, low-gear road climbing, and low-gear field driving. The position of the two-position three-way valve, part of the gear-shifting and breaking valve assembly, was controlled by the input control signal to achieve gear switching, and different load values were input to correspond to different operating conditions. The input load values and control signals for each group of motors in the hydraulic system are listed in Table 8.

## Analysis of simulation results

The actual driving of the wide-span farming platform was divided into three stages: start-up acceleration, constant-speed driving, and deceleration to stop. Based on the control characteristics graph of the selected variable-displacement pump, the control current of the variable-displacement pump is shown in Fig 9.

The corresponding parameters and input signals of the rotary friction torque generator were set according to the input load values given in Fig 10. Simulation results of the variable-displacement pump and the hydraulic motor under steady road driving conditions were obtained, as shown in Figs 10 and 11. During driving, the outlet pressure of the variable pump rapidly rose within the initial 0.3 s of the start-up phase, followed by oscillations for the next

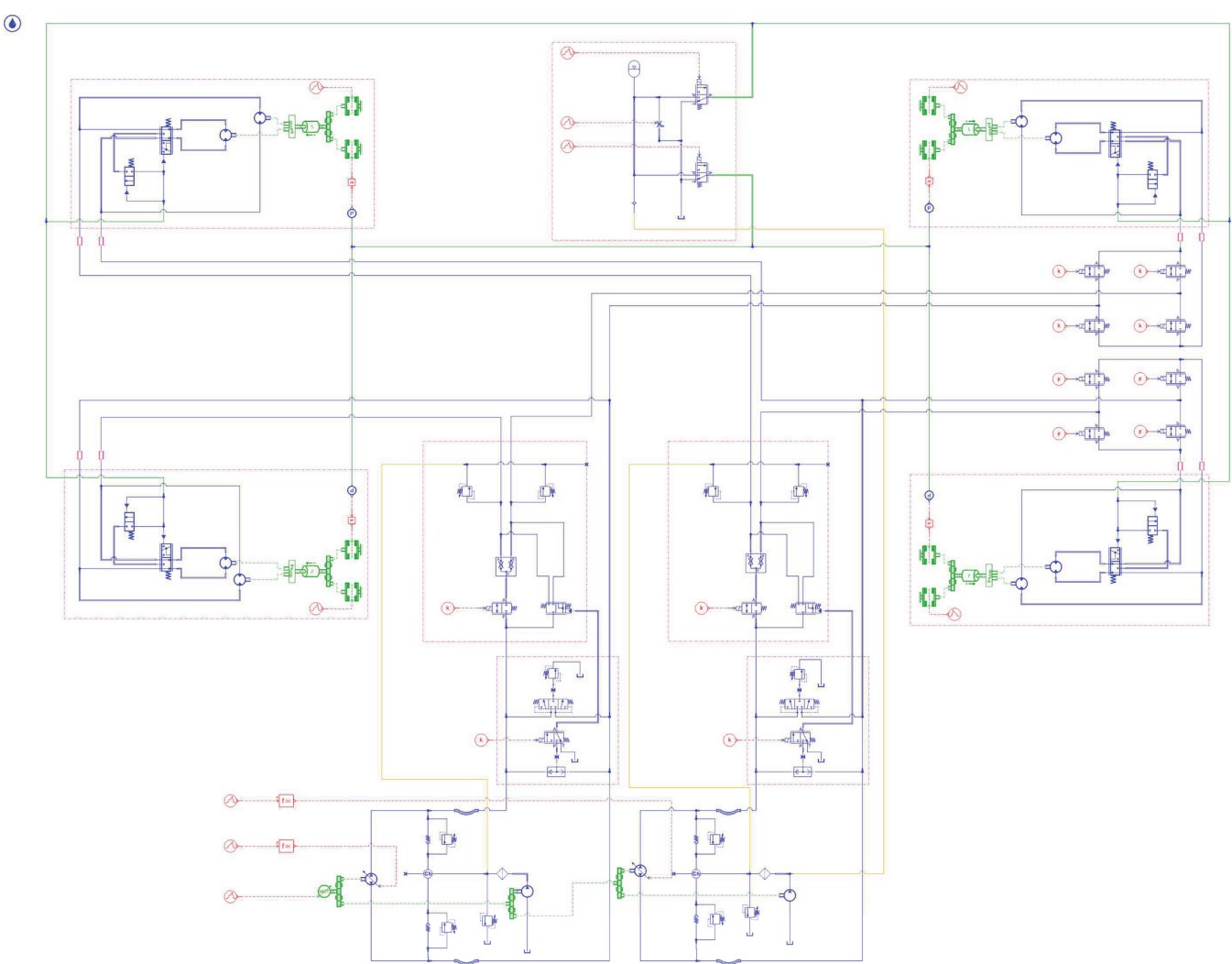

**Fig 8. AMESim simulation model of the hydraulic drive system.**

Table 7. Main simulation parameters of the hydraulic drive system simulation model.

| Main Parameter | Numerical Value |
|---|---|
| Engine speed (r/min) | 2200 |
| Variable pump displacement (mL/r) | 165 |
| Hydraulic motor displacement (mL/r) | 1050 |
| Directional valve control current (mA) | 40 |
| Hydraulic pipeline length (m) | 4 |

Table 8. Input values of each hydraulic motor group.

| Driving Condition | Control Signal (mA) | Input Load (N·m) |
|---|---|---|
| Steady road driving | 40 | 4000 |
| Road climbing | 0 | 9500 |
| Field driving | 0 | 7200 |

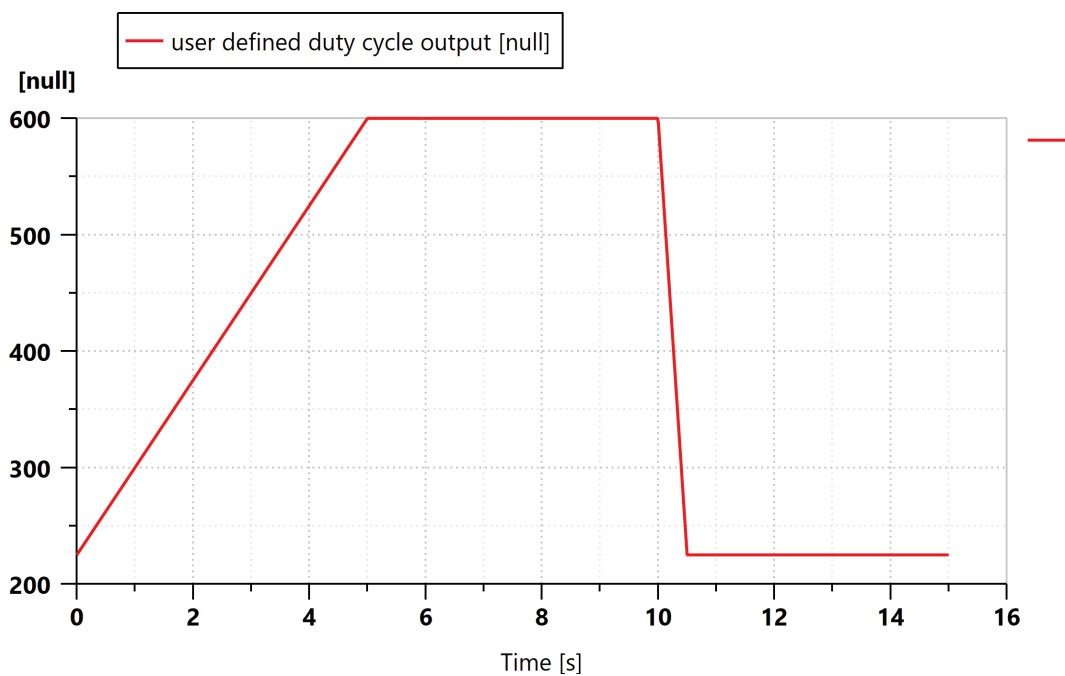

**Fig 9. Control current of the variable-displacement pump.** The control current of the variable pump increased from 225 to 600 mA within 5 s, causing the displacement ratio of the variable pump to gradually increase from 0 to 1. It remained at the maximum displacement from 5 to 10 s. At 10 s, the deceleration began, and the variable-pump displacement decreased to 0 within 0.5 s.

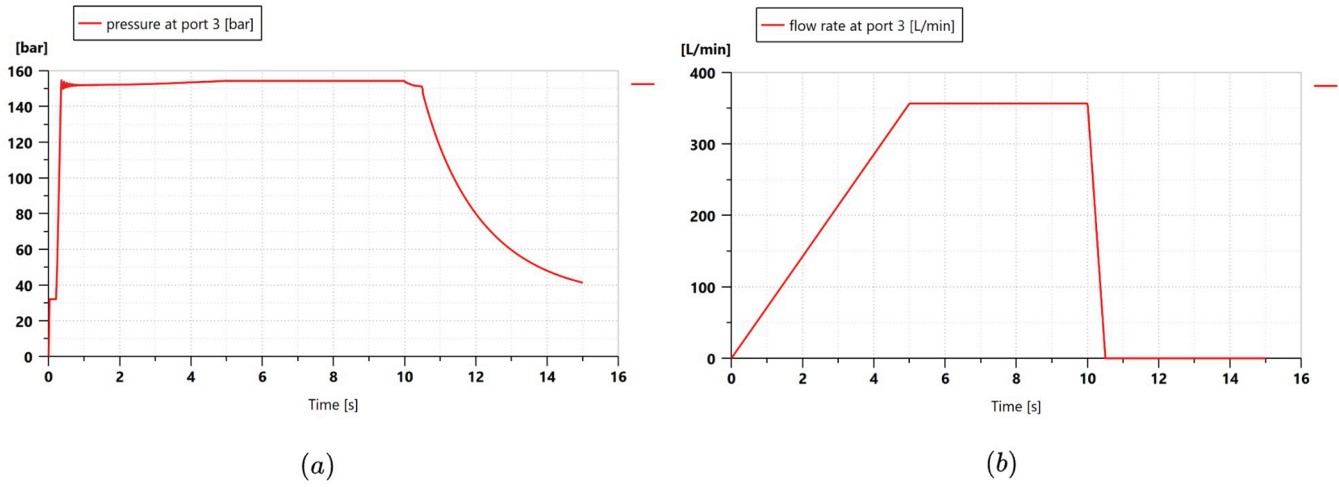

**Fig 10. Simulation results of the variable-displacement pump under steady road driving conditions.** a: Outlet pressure. b: Output flow rate.

0.5 s. The outlet pressure gradually increased during the acceleration phase, reaching a maximum value of 15.4 MPa at 5 s, and it remained stable during the 5 to 10 s of constant-speed driving. After deceleration began, the outlet pressure decreased accordingly. The output flow rate changed consistently with the variable pump control signal, with the maximum value stabilizing at 356.8 L/min.

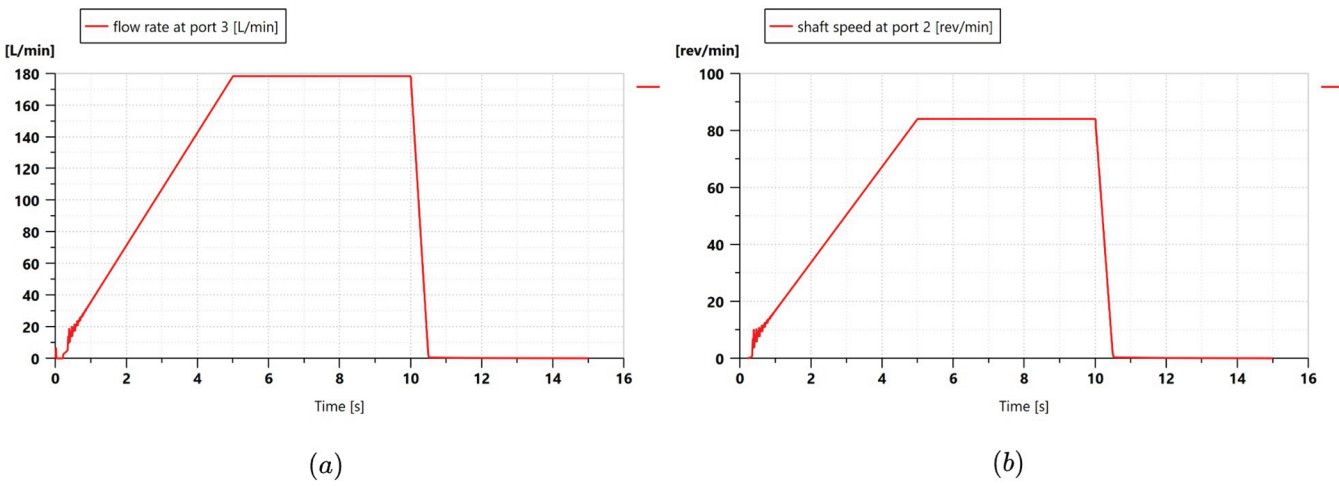

**Fig 11. Simulation results of the hydraulic motor under steady road driving conditions.** a: Flow rate. b: Speed.

According to the analysis of Fig 11, 0.3 s after start-up, the hydraulic oil rushed into the motor, causing an impact and a sudden change in the flow rate within a short period, accompanied by oscillations. Subsequently, the flow rate stabilized, and the variation trend of the rotational speed aligned with the variation trend of the flow rate. The maximum flow rate was 178.4 L/min, and the maximum speed was 84 r/min. The output torque of the hydraulic motor stabilized after the oscillation ended and decreased after the rotational speed dropped to zero.

Additionally, in the longitudinal mode in a low gear under road climbing conditions, the maximum flow rate was 157 L/min, and the maximum pressure was 15.3 MPa. In the transverse mode in a low gear under field driving conditions, the maximum flow rate was 173 L/min and the maximum pressure was 18.3 MPa. The simulation result curves under these two driving conditions exhibit similar trends to those shown in Figs 10 and 11, and are therefore not presented here.

## Experiments and analysis

Experiments on the driving system of the wide-span platform were conducted to study the platform's driving performance under different modes and driving conditions. Driving performance and hydraulic characteristic tests of the hydraulic drive system were performed. The wide-span platform in the two driving modes is shown in Fig 12.

The controller of the hydraulic electro-control system for the tracked wide-span farming platform directly outputs a PWM signal to regulate the flow rate of the variable-displacement pump, thereby controlling the travel speed. Simultaneously, to achieve steering control, the opening and direction of the hydraulic steering proportional valve are controlled based on the PID principle. The wide-span farming platform can be operated by the driver using a joystick located in the cab or through a wireless remote control. The measurement system configuration for parameter acquisition in the wide-span farming platform is shown in Fig 13. To ascertain the hydraulic performance characteristics of the platform, four flow sensors were installed within the hydraulic drive system's circuit. Specifically, a flow sensor was positioned at the oil inlet pipeline of each track hydraulic motor during the platform's forward motion in different driving modes. A pressure sensor was also mounted adjacent to the flow sensor to capture

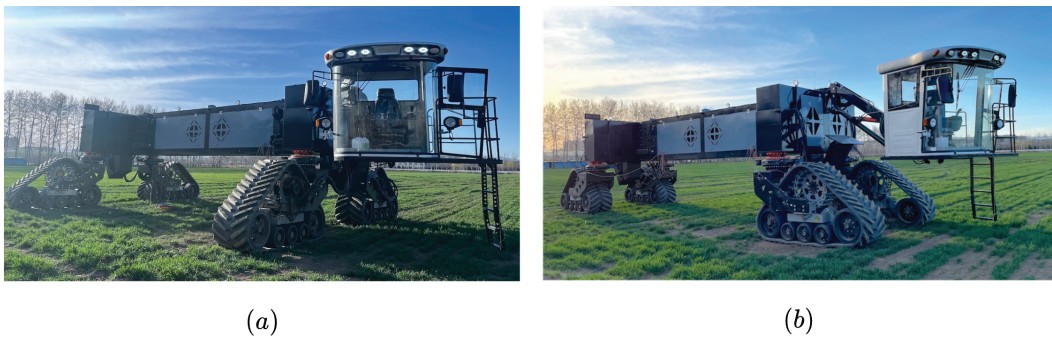

$(a)$ $(b)$

**Fig 12. Wide-span farming platform.** (a) Transverse driving mode. (b) Longitudinal driving mode.

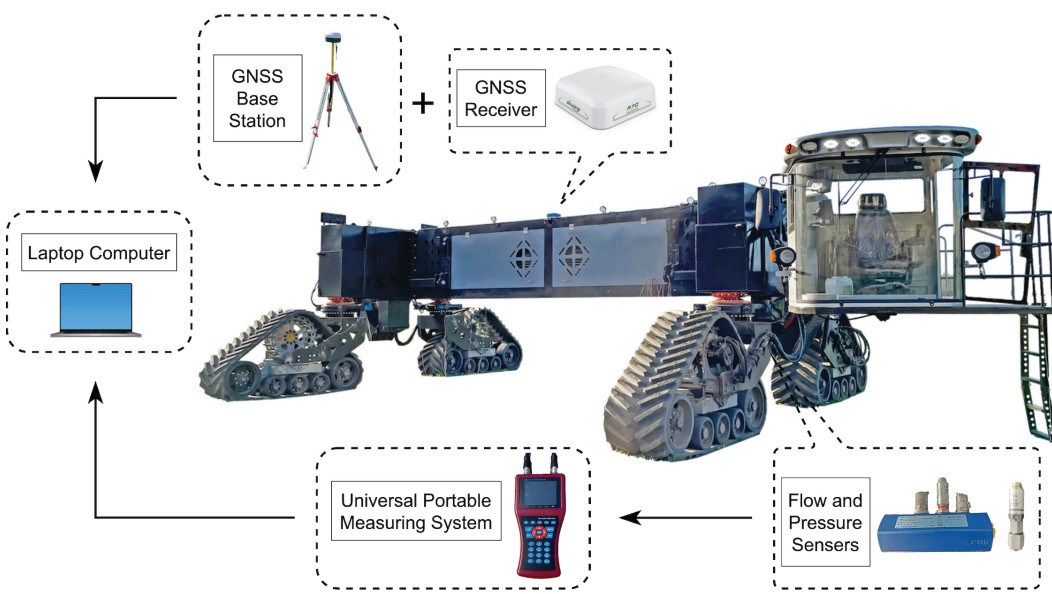

**Fig 13. Measurement system configuration for parameter acquisition in the wide-span farming platform.**

the pressure and flow data simultaneously. This data was subsequently read by a measuring instrument for two diagonally positioned track hydraulic motors, both of which were actuated by the same variable pump. A global navigation satellite system (GNSS) receiver was used to record the platform's field travel speed and trajectory data, with the GNSS base station.

## Driving performance

The deviation rate is the ratio of the lateral deviation of the chassis from the reference straight line to the longitudinal travel distance. It is an important indicator for evaluating the driving performances of agricultural tractors. A lower deviation rate contributes to improved chassis maneuverability and enhances row efficiency during field operations. It also helps reduce the crop trampling rate, thereby increasing the operational efficiency of the tracked wide-span platform and improving crop yields. With the longitudinal mode as an example, the schematic diagram of the platform's driving deviation is shown in Fig 14. The same principle applies to the transverse mode.

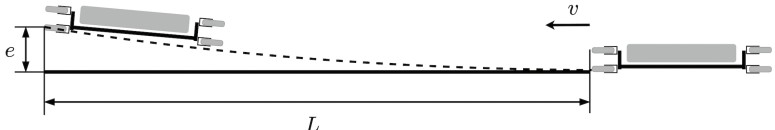

**Fig 14. Schematic diagram of the platform's driving deviation.**

Therefore, according to Fig 14, the deviation rate can be calculated by the following equation:

$$p = \frac{e}{L} \times 100\% \qquad (12)$$

where $e$ is the lateral deviation, and $L$ is the theoretical straight-line travel distance.

**Longitudinal driving mode.** Since the tracked wide-span farming platform primarily operates in a longitudinal orientation on hard concrete surfaces, the deviation rate test of the platform was first conducted on level, hard concrete ground. At the commencement of the measurement, the front end of the front track of the farming platform was aligned with the starting line. The platform then shifted into a low gear, maintaining a straight course without steering adjustments until any track reached the deceleration line. At this point, remote control was used to cease acceleration, allowing the track to decelerate to a stop.

The deviation rate test results for the longitudinal driving mode on hard cement pavement, as derived from measurements and calculations, are presented in Table 9. The data indicated that when the platform was traversing a horizontal hard surface, the average deviation rate in longitudinal mode was 5.0%. According to the national standard GB/T 15370.4-2012 "General Technical Conditions for Agricultural Tractors—Part 4: Crawler Tractors," the deflection rate for tractors should not exceed 6% on dry, flat road surfaces with slopes of less than 1% (both vertically and horizontally). As there is currently no existing standard for the deviation rate of tracked wide-span farming platforms, the evaluation is conducted based on the standard for tracked tractors. However, due to the use of large semi-tracked wheels and the wide-span structural design, the actual deviation rate of the tracked wide-span platform is higher than that of conventional tracked tractors. Consequently, the deviation rate of the tracked wide-span farming platform in longitudinal mode largely complied with the stipulations of this national standard.

**Table 9. Deviation rate test results for the longitudinal driving mode.**

| Group | Engine Speed (r/min) | Time (s) | Average Velocity (km/h) | Deviation (m) | Travel Distance (m) | Deviation Rate (%) | Average Deviation Rate (%) |
|---|---|---|---|---|---|---|---|
| 1 | 1200 | 27.0 | 6.67 | 2.75 | 50.87 | 5.4 | 5.0 |
| 2 | | 27.2 | 6.62 | 2.28 | 50.59 | 4.5 | |
| 3 | 1500 | 21.0 | 8.57 | 2.62 | 51.14 | 5.1 | |
| 4 | | 20.7 | 8.70 | 2.67 | 51.35 | 5.2 | |
| 5 | 1800 | 18.3 | 9.84 | 2.71 | 52.16 | 5.2 | |
| 6 | | 18.5 | 9.73 | 2.80 | 51.94 | 5.4 | |
| 7 | 2000 | 15.4 | 11.69 | 2.42 | 53.20 | 4.5 | |
| 8 | | 15.9 | 11.32 | 2.58 | 53.17 | 4.9 | |

**Transverse driving mode.** The wide-span farming platform was driven on soft soil farmland in transverse driving mode. The starting point of the platform's travel and the endpoint were marked, which corresponded to a straight-line distance of 150 m. The engine speed of the platform was set to 2200 r/min, and it started to accelerate from the starting point in a low gear. After reaching the maximum speed, it traveled at a constant speed for a period, after which it decelerated to a stop near the endpoint. The platform's field traveling speed and trajectory data were recorded using the GNSS receiver. The raw data collected was processed using MATLAB, and the travel deviation and speed changes of the platform in transverse mode under dry and wet soil conditions are shown in Figs 15 and 16. The movmean filtering method was applied to process the data and reduce random noise, thereby smoothing the signal curves.

Based on the results from Figs 15 and 16, the maximum travel velocity of the platform in transverse mode was 12.3 km/h, with a deviation rate of 6.0% on dry soil and 2.1% on wet soil. The deviation rate and maximum travel speed on wet soil were both lower than those on dry soil. This was attributed to the greater resistance to travel and steering in wet soil, which hindered the travel deviation motion of the semi-tracks.

## Driving hydraulic characteristics

**Longitudinal driving mode.** In this test, the wide-span farming platform operated on a horizontal cement hard surface, moving in a straight line in longitudinal mode. It started to accelerate at about 15 s, traveled at maximum speed for a specified distance, and then decelerated to a stop at the 47.5 m deceleration mark. The flow and pressure changes of the two diagonally distributed track hydraulic motors driven by the same pump were measured over a period of 60 s. The engine speed was adjusted to 2000 r/min. The driving gear was set to high and low in separate tests. The hydraulic characteristics test results for the track hydraulic motors of the platform traveling longitudinally on the road are shown in Figs 17 and 18.

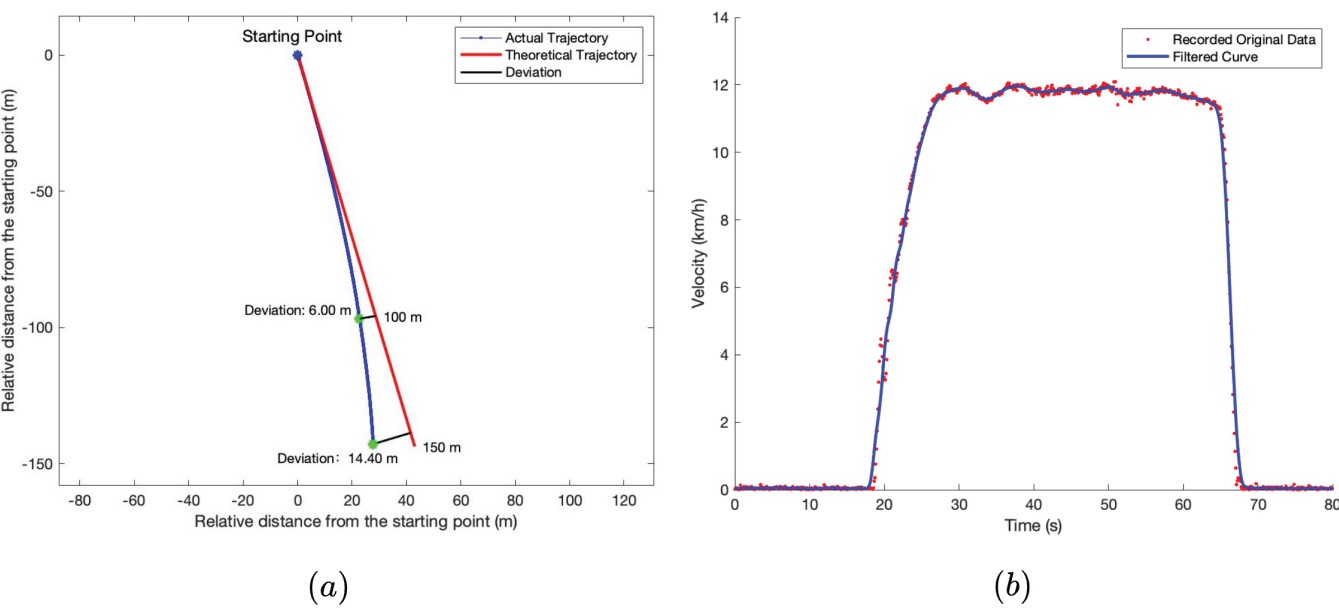

$(a)$ $(b)$

**Fig 15. Driving performance test results of the wide-span farming platform on dry, soft soil farmland in transverse driving mode.** a: Farmland traveling deviation trajectory. b: Farmland traveling velocity variation.

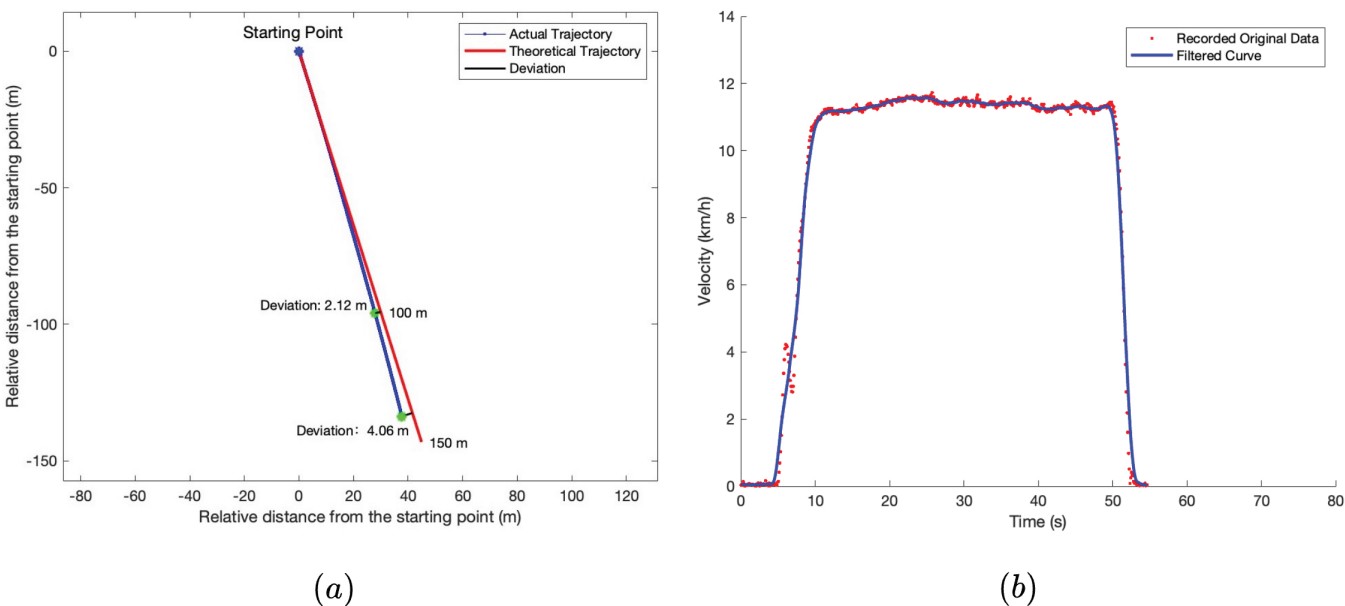

$(a)$                                                                                                           $(b)$

**Fig 16. Driving performance test results of the wide-span farming platform on wet, soft soil farmland in transverse driving mode.** a: Farmland traveling deviation trajectory. b: Farmland traveling velocity variation.

The relevant data for the hydraulic characteristics of the platform's straight-line driving on the road in longitudinal mode are shown in Table 10.

According to Figs 17 and 18, the platform's driving process can be divided into three stages:

- Stage 1: In the acceleration stage from a standstill, the pressure of the track hydraulic motor rose steeply, indicating that the driving force of the platform was at the maximum. At the same time, the flow rate of the track hydraulic motor gradually increased as the platform's speed rose, and the pressure began to decrease.
- Stage 2: In the constant-speed stage, the pressure and flow rate of the track hydraulic motor became stable and remain unchanged.
- Stage 3: In the braking to stopping stage, the flow rate of the track hydraulic motor rapidly dropped to zero, and the pressure decreased to the system pressure.

When the platform's engine speed was 2000 r/min and the platform operated in longitudinal mode on the road, the hydraulic characteristic results in high and low gears were as follows. In a high gear, the dual-displacement motor operated in half-displacement mode. The maximum acceleration pressure during the acceleration stage from still was 452 bar, and during the constant-speed stage, the pressure stabilized at around 229 bar, with a constant flow rate of 115 L/min. In a low gear, the dual-displacement motor switched to full-displacement mode to provide a greater driving force when driving at a constant speed. The maximum acceleration pressure during the acceleration stage was 357 bar, and during the constant-speed stage, the pressure stabilized at around 158 bar, with a constant flow rate of 162 L/min.

**Transverse driving mode.** In this test, the wide-span farming platform operated on soft soil in a field, moving in a straight line in transverse mode. The remote control handle was operated to accelerate the platform from a stationary state. After reaching the maximum

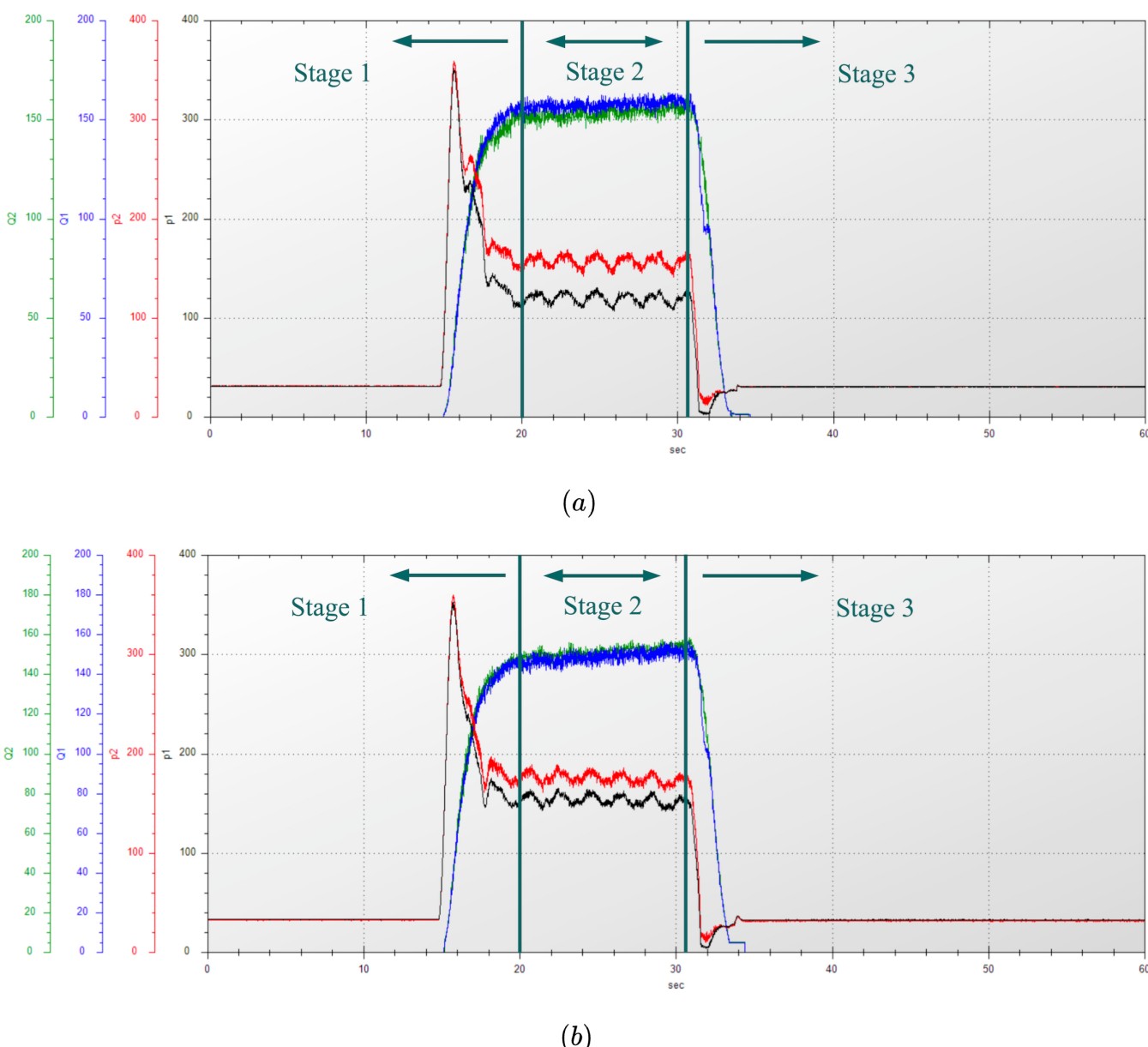

**Fig 17. Variation of track hydraulic motor pressure and flow rate at engine speed of 2000 r/min in a low gear traveling longitudinally on road.** (a) First motor–pump system. (b) Second motor–pump system. Q1 and P1 represent the flow rate and pressure of the front tracks, while Q2 and P2 represent those of the rear tracks.

speed, a constant speed was maintained for a certain distance, then acceleration was stopped, and the platform was decelerated to a stop. The engine speed was set to 2200 r/min and the gear was set to low. The difference in the pressure trends of the hydraulic motors on both sides of the platform is the result of their being a part of different motor-pump systems. The hydraulic characteristic test results of the track hydraulic motor when the platform was in transverse mode on soft soil farmland are shown in Fig 19. The relevant data for the hydraulic characteristics of the platform traveling in transverse mode on soft soil farmland are shown in Table 11.

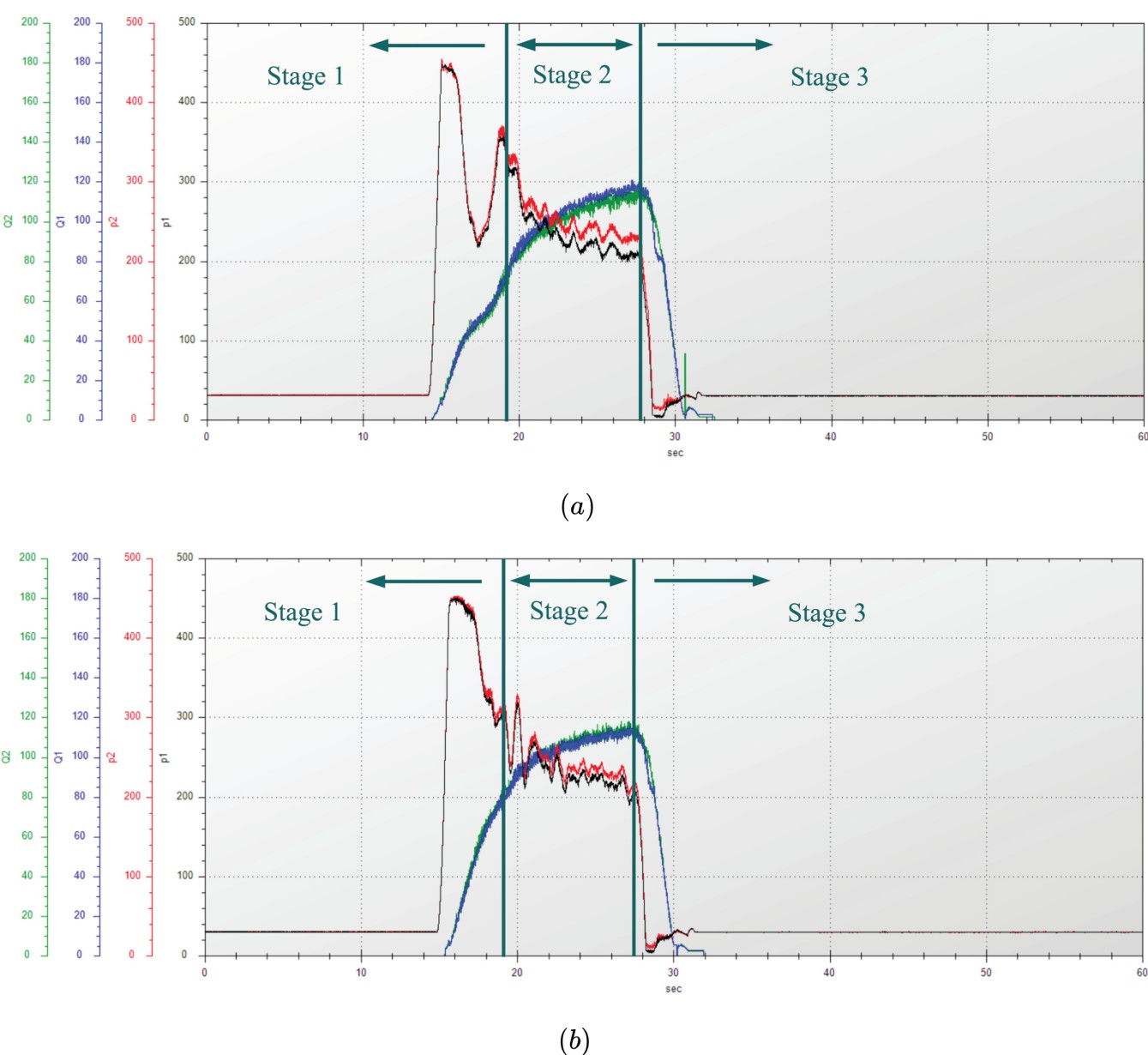

**Fig 18. Variation of track hydraulic motor pressure and flow rate at engine speed of 2000 r/min in a high gear traveling longitudinally on road.** (a) First motor–pump system. (b) Second motor–pump system. Q1 and P1 represent the flow rate and pressure of the front tracks, while Q2 and P2 represent those of the rear tracks.

**Table 10. Relevant data for the hydraulic characteristics of the platform driving in a straight line on a road in longitudinal mode.**

| Engine Speed (r/min) | Gear | Constant Flow Rate (L/min) | Maximum Acceleration Pressure (bar) | Constant Acceleration Pressure (bar) |
|---|---|---|---|---|
| 2000 | Low | 162 | 357 | 158 |
|  | High | 115 | 452 | 229 |

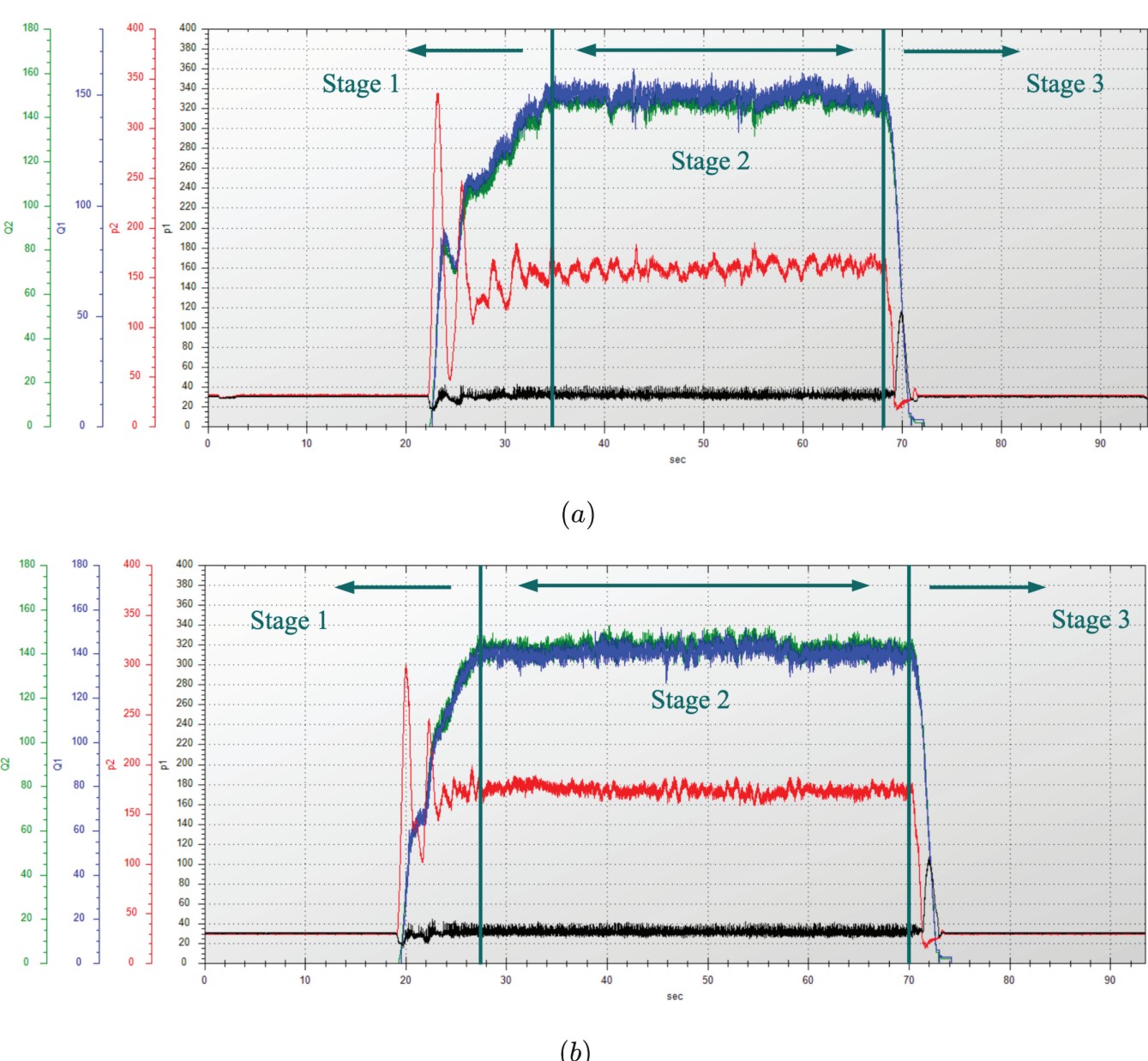

**Fig 19. Flow rate and pressure changes of the track hydraulic motors on one side of the platform traveling in transverse mode on soft soil farmland.**
(a) Dry, soft soil. (b) Wet, soft soil.

**Table 11.** Relevant data for the hydraulic characteristics of the platform traveling in transverse mode on soft soil farmland.

| Engine Speed (r/min) | Gear | Soil Condition | Constant Flow Rate (L/min) | Maximum Acceleration Pressure (bar) | Constant Acceleration Pressure (bar) |
|---|---|---|---|---|---|
| 2200 | Low | Dry | 150 | 334 | 164 |
| | | Wet | 145 | 298 | 175 |

When the platform's engine speed was 2200 r/min and the platform operated in transverse mode in a low gear on the farmland, the hydraulic characteristic results under dry and wet soil conditions were as follows. On dry, soft soil, the maximum acceleration pressure during the acceleration stage from still was 334 bar, and during the constant-speed stage, the pressure stabilized at around 164 bar, with a constant flow rate of 150 L/min. On wet, soft soil, the dual-displacement motor provided a greater driving force when driving at a constant speed. The maximum acceleration pressure during the acceleration stage was 298 bar, and during the constant-speed stage, the pressure stabilized at around 175 bar, with a constant flow rate of 145 L/min. The reasons for the smaller lateral deviation under softer, muddy conditions are as follows: compared with dry soil, the coefficient of terrain deformation resistance is higher and the sinkage beneath the track–ground contact patch is greater. While the available tractive effort increases, the heaved soil berms along both sides of the track impose greater lateral resistance, thereby inhibiting the platform's drift.

## Conclusion

### Hydraulic drive system based on multiple driving modes

The wide-span farming platform has two basic driving modes: longitudinal and transverse, as well as a special rotational steering mode. Therefore, the hydraulic drive system of such a chassis needs to support mode-switching functionality, enabling walking and steering in different modes while ensuring system stability across various road surfaces and operational scenarios. In designing the hydraulic drive system, changes in the hydraulic characteristics with different walking modes, as well as variations in the output speeds and torques of the hydraulic motors, must be considered. Additionally, the movement matching relationship between all actuators (hydraulic motors) after a mode switch is essential to ensure system synchronization and stability. Thus, an X-type dual-pump four-motor system was designed in this study, where two closed-loop piston pumps drive four independently distributed dual-displacement hydraulic motors. The synchronization and stability are improved through the use of a proportion diverter valve and the X-type pump–motor distribution system. The motor rotation control valve group is used to change the rotation direction of the hydraulic motors to adapt to different walking modes. This provides insight and a reference for the design of a hydraulic drive system capable of multiple walking modes, as well as for the structural and control system algorithm optimization of the platform.

### Driving performance of chassis with wide-span structure

For the asymmetric, wide-span U-shaped structure design of the platform, the position of the center of mass significantly affects the force in the vertical direction between the walking mechanism and the ground, resulting in differences in the driving force and causing deviations in movement. By employing an X-type dual-pump four-motor system, control over the deviation rate was improved, enhancing the stability and meeting the driving requirements. However, in practice, the primary cause of deviation in the platform is mechanical error which are installation tolerances and neutral-position calibration errors of the track modules. In addition, because the platform has a wide-span gantry, the hydraulic pipelines are long and asymmetric, resulting in unequal pressure losses; as the oil temperature rises, hydrostatic losses increase and efficiency decreases, leading to unequal actual output flow rates of the hydraulic motors on the two sides. There are deviations of the speed and torque between these motors and thereby exacerbating the platform's deviation. The average deviation rate of the platform in longitudinal mode on cement roads was 5.0%. The deviations of

the platform in transverse mode on dry and wet, soft soil at the maximum driving speed were 6% and 2.1%, respectively. Analysis of the experimental results identified several key factors impacting the performance of the wide-span platform:

- Hydraulic system leakage and losses,
- Installation and calibration errors in the walking mechanisms, such as tires or tracks,
- Center of mass offset of the platform,
- Different driving modes: longitudinal and transverse,
- Surface characteristics (e.g., asphalt, cement road, and soft soil) and operating conditions (e.g., plowing and planting).

In a future study, we will consider co-simulation of the hydraulic system with a multi-body dynamics model. We will drive the virtual prototype using the established hydraulic simulation, and assess the deviation rate under simulated terrain conditions. The results will be compared against field measurements of relevant parameters to elucidate how component settings relate to the observed deviation. We will focus on the four-wheel independent steering hydraulic system and a four-wheel independent electric-drive system. With steering-angle and wheel-speed sensors, we will develop an electronic differential with variable wheel track (axle spacing) to enforce the kinematic relationship between wheel angle and rotational speed, thereby achieving precise closed-loop control of steering angle and wheel velocity, further enhancing the stability, flexibility, and synchronization of the wide-span farming platform's driving system.

## Patents

The following patent has been granted in connection with the innovative aspects of the research presented in this manuscript:

- Patent Name: "An Agricultural Machinery Operation Platform"
- Inventors: Xianfa Fang, Haihua Wu, Qing He, Hongfeng Yan, Wenke Chen, Tian Tian, Shuanglong Jia, Yongliang Zhao, Mingshuai Yuan, Qi Zhang, Chengli Liu, and Jiaxiang Ma
- Patent Number: CN 216994593 U
- Patent Office: China National Intellectual Property Administration (CNIPA)
- Status: Granted on [2022.07.19], valid in participating Chinese countries

## Supporting information

**S1 Text. This is the raw data from the performance test of the platform, obtained through GNSS measurements under the transverse driving mode on dry, soft soil farmland.**
(TXT)

**S2 Text. This is the raw data from the performance test of the platform, obtained through GNSS measurements under the transverse driving mode on wet, soft soil farmland.**
(TXT)

**S3 Text. This is the MATLAB file used to process the acquired GNSS data and generate the corresponding figures (Figs 15 and 16).**
(M)

## Author contributions

**Conceptualization:** Hongfeng Yan, Xianfa Fang.

**Data curation:** Rongxuan Li, Falian Li.

**Formal analysis:** Jiaxiang Ma, Rongxuan Li, Falian Li.

**Funding acquisition:** Hongfeng Yan, Xianfa Fang.

**Investigation:** Hongfeng Yan, Rongxuan Li, Xianfa Fang.

**Methodology:** Jiaxiang Ma, Hongfeng Yan, Xianfa Fang.

**Project administration:** Hongfeng Yan, Xianfa Fang.

**Resources:** Jiaxiang Ma, Falian Li.

**Software:** Jiaxiang Ma, Falian Li.

**Supervision:** Hongfeng Yan, Rongxuan Li, Falian Li, Xianfa Fang.

**Validation:** Hongfeng Yan, Rongxuan Li.

**Visualization:** Jiaxiang Ma, Rongxuan Li.

**Writing – original draft:** Jiaxiang Ma, Hongfeng Yan.

**Writing – review & editing:** Jiaxiang Ma, Hongfeng Yan, Falian Li.

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
