## [Decision Letter · Decision Letter 0]

11 Aug 2025

PONE-D-25-23565Design, simulation, and experimental study of hydrostatic drive system for wide-span farming platformPLOS ONE

Dear Dr. Yan,

Thank you for submitting your manuscript to PLOS ONE. After careful consideration, we feel that it has merit but does not fully meet PLOS ONE’s publication criteria as it currently stands. Therefore, we invite you to submit a revised version of the manuscript that addresses the points raised during the review process.

We look forward to receiving your revised manuscript.

Kind regards,

sunny narayan

Academic Editor

PLOS ONE

Journal Requirements:

[This research was funded by Research Project of China National Machinery Industry Corporation(ZDZX2022-1).]. 

For example, authors should submit the following data

4. Please note that PLOS ONE has specific guidelines on code sharing for submissions in which author-generated code underpins the findings in the manuscript. In these cases, we expect all author-generated code to be made available without restrictions upon publication of the work. Please review our guidelines at https://journals.plos.org/plosone/s/materials-and-software-sharing#loc-sharing-code and ensure that your code is shared in a way that follows best practice and facilitates reproducibility and reuse.

Reviewers' comments:

Reviewer's Responses to Questions

**Comments to the Author**

1. Is the manuscript technically sound, and do the data support the conclusions?

Reviewer #1: Yes

2. Has the statistical analysis been performed appropriately and rigorously? 

Reviewer #1: N/A

3. Have the authors made all data underlying the findings in their manuscript fully available?

Reviewer #1: Yes

4. Is the manuscript presented in an intelligible fashion and written in standard English?

Reviewer #1: Yes

5. Review Comments to the Author

Reviewer #1: I am treating the article as a report of performance of a working prototype of wide span farming platform using hydrostatic drive which is different from a research paper. The report covers the hardware description and the AMESim simulation of the hydraulic system, which in turn gives the simulated components performance in terms of flowrate, pressure etc. which can be verified with measurement of flow within the actual circuit.

The parameters not reported and needed to be further explained is what causes the deviation and how this is tied up with the flow parameters in the circuit. Is the individual hydraulic motor rpm and torque monitored (motor 1 ,2, 3 and 4)? Are there deviations of the speed and torque between these motors (maybe due to slip in one of the track?). What about the weight distribution and the centre of gravity in each transverse and longitudinal arrangement. Noticeably the softer mud/earth condition gives less deviation - indicating the variabiity of traction (friction coefficient?).

Would a control with feedback eg steering angle parameter matching the individual motor speed etc) which can further explain the dynamic of the wide span farming platform developed here. I do hope the team can better explain and highlight the relationship between the AMesim component parameters and the deviation obtained. A dynamic model of the platform with adequate degrees of freedom and the individual traction and with trajectory prediction as a function of individual traction can be used to achieve this.

6. PLOS authors have the option to publish the peer review history of their article (what does this mean?). If published, this will include your full peer review and any attached files.

Reviewer #1: No

---

## [Author Response · Author response to Decision Letter 1]

12 Sep 2025

Dear editor and reviewer,

Thank you for offering us an opportunity to improve the quality of our submitted manuscript (Design, simulation, and experimental study of hydrostatic drive system for wide-span farming platform). We appreciated very much the reviewer’s constructive and insightful comments. In this revision, we have addressed all of these comments/suggestions. We hope the revised manuscript has now met the publication standard of your journal.

In the following, we include a point-by-point response to the comments from the reviewer. In the revised manuscript, all the changes have been highlighted in red.

Reviewer #1

Comment 1: The parameters not reported and needed to be further explained is what causes the deviation and how this is tied up with the flow parameters in the circuit.

Response: Due to the platform’s uneven weight distribution, the tracks on the heavier side experience greater ground resistance. The hydraulic system then preferentially allocates flow to the lower-resistance side, which tends to induce deviation during straight-line travel. To address this, we incorporated the proportion diverter valve to enforce equal flow to both sides in straight-ahead operation, thereby ensuring straight-line performance. However, in practice, the primary cause of deviation in the platform is mechanical error which are installation tolerances and neutral-position calibration errors of the track modules. In addition, because the platform has a wide-span gantry, the hydraulic pipelines are long and asymmetric, resulting in unequal pressure losses; as the oil temperature rises, hydrostatic losses increase and efficiency decreases, leading to unequal actual output flow rates of the hydraulic motors on the two sides. There are deviations of the speed and torque between these motors and thereby exacerbating the platform’s deviation. We have added these explanations in page 24, conclusion part.

Comment 2: Is the individual hydraulic motor rpm and torque monitored (motor 1 ,2, 3 and 4)?

Response: Yes. We installed encoders coaxial with the drive sprockets of the track modules to measure the actual output speed. Because the hydraulic motor torque is proportional to the pressure differential across the motor, it can be inferred by measuring the inlet–outlet pressures and computing the differential. Accordingly, we monitored only the pressures of each hydraulic motor.

Comment 3: Are there deviations of the speed and torque between these motors (maybe due to slip in one of the track?).

Response: Yes, there are. They arise primarily from track slip and hydrostatic losses in the hydraulic system. During turning, the discrepancies are mainly due to the hydrostatic system’s self-adaptive flow distribution characteristics. We have added these explanations in page 24, conclusion part.

Comment 4: What about the weight distribution and the centre of gravity in each transverse and longitudinal arrangement.

Response: Thanks for your suggestion. We have added the Fig 3 to describe the center of gravity in each transverse and longitudinal arrangement

Fig 3. Overall dimensions of the wide-span farming platform. (a) Longitudinal mode. (b) Transverse mode.

Comment 5: Noticeably the softer mud/earth condition gives less deviation - indicating the variability of traction (friction coefficient?).

Response: The reasons for the smaller lateral deviation under softer, muddy conditions are as follows: compared with dry soil, the coefficient of terrain deformation resistance is higher and the sinkage beneath the track–ground contact patch is greater. While the available tractive effort increases, the heaved soil berms along both sides of the track impose greater lateral resistance, thereby inhibiting the platform’s deviation. We have added these explanations in page 23, driving hydraulic characteristics part.

Comment 6: Would a control with feedback eg steering angle parameter matching the individual motor speed etc) which can further explain the dynamic of the wide span farming platform developed here.

Response: Thanks for your advice. We are indeed conducting related work. In this generation of the platform, we adopted a diagonal flow-splitting control scheme in which two variable-displacement pumps independently drive the hydraulics motors arranged on opposite corners of the chassis. In both straight-line travel and turning, the closed hydraulic circuit automatically apportions differential motor speeds, enabling stepless speed regulation and smooth operation as verified experimentally. In subsequent work, leveraging the platform’s structural characteristics, we will design a four-wheel independent steering hydraulic system and a four-wheel independent electric-drive system. With steering-angle and wheel-speed sensors, we will develop an electronic differential with variable wheel track (axle spacing) to enforce the kinematic relationship between wheel angle and rotational speed, thereby achieving precise closed-loop control of steering angle and wheel velocity. We have added these explanations in page 25, conclusion part.

Comment 7: I do hope the team can better explain and highlight the relationship between the AMESim component parameters and the deviation obtained. A dynamic model of the platform with adequate degrees of freedom and the individual traction and with trajectory prediction as a function of individual traction can be used to achieve this.

Response: The proportion diverter valve ensures identical inlet flow to the track hydraulic motors on both sides of the platform, which is fundamental to maintaining straight-line tracking performance. However, the specific lateral deviation behavior must be evaluated under real ground conditions. In future work, we will consider co-simulation of the hydraulic system with a multibody dynamics model. We will drive the virtual prototype using the established hydraulic simulation, then assess the deviation rate under simulated terrain conditions. The results will be compared against field measurements of relevant parameters to elucidate how component settings relate to the observed deviation. We have added these explanations in page 25, conclusion part.

---

## [Editor Report · Decision Letter 1]

2 Oct 2025

Design, simulation, and experimental study of hydrostatic drive system for wide-span farming platform

PONE-D-25-23565R1

Dear Dr. 

We’re pleased to inform you that your manuscript has been judged scientifically suitable for publication and will be formally accepted for publication once it meets all outstanding technical requirements.

Kind regards,

sunny narayan

Academic Editor

PLOS ONE
---

## [Editor Report · Acceptance letter]

PONE-D-25-23565R1

PLOS ONE

Dear Dr. Yan,

I'm pleased to inform you that your manuscript has been deemed suitable for publication in PLOS ONE. Congratulations! Your manuscript is now being handed over to our production team.

Kind regards,

on behalf of

Dr. sunny narayan

Academic Editor

PLOS ONE